# *Brucella* effectors NyxA and NyxB target SENP3 to modulate the subcellular localisation of nucleolar proteins

Arthur Louche[1,9], Amandine Blanco[1,9], Thais Lourdes Santos Lacerda[1], Lison Cancade-Veyre[1], Claire Lionnet[2], Célia Bergé[1], Monica Rolando[3], Frédérique Lembo[4], Jean-Paul Borg [4,5], Carmen Buchrieser [3], Masami Nagahama[6], Francine C. A. Gérard[1], Jean-Pierre Gorvel [7], Virginie Gueguen-Chaignon[8], Laurent Terradot [1] ✉ & Suzana P. Salcedo [1] ✉

The cell nucleus is a primary target for intracellular bacterial pathogens to counteract immune responses and hijack host signalling pathways to cause disease. Here we identify two *Brucella abortus* effectors, NyxA and NyxB, that interfere with host protease SENP3, and this facilitates intracellular replication of the pathogen. The translocated Nyx effectors directly interact with SENP3 via a defined acidic patch (identified from the crystal structure of NyxB), preventing nucleolar localisation of SENP3 at late stages of infection. By sequestering SENP3, the effectors promote cytoplasmic accumulation of nucleolar AAA-ATPase NVL and ribosomal protein L5 (RPL5) in effector-enriched structures in the vicinity of replicating bacteria. The shuttling of ribosomal biogenesis-associated nucleolar proteins is inhibited by SENP3 and requires the autophagy-initiation protein Beclin1 and the SUMO-E3 ligase PIAS3. Our results highlight a nucleomodulatory function of two *Brucella* effectors and reveal that SENP3 is a crucial regulator of the subcellular localisation of nucleolar proteins during *Brucella* infection, promoting intracellular replication of the pathogen.

The spatial organisation of eukaryotic cells and accurate subcellular targeting of macromolecules are essential for the regulation of numerous cellular processes, namely cell growth, survival and stress responses. Indeed, the mislocalisation of proteins in the cell has been associated with cellular stress and multiple diseases[1]. The nucleus plays a critical role in sensing and orchestrating the overall host responses to stress. Several nuclear proteins have been shown to act as important stress sensors in the cell, including the sentrin-specific protease 3 (SENP3), which is also involved in the regulation of the cell cycle, survival pathways and ribosomal biogenesis[2]. SENP3 functions as a processing enzyme for the SUMO2/3 precursor and as a deconjugase of SUMOylated substrates. SENP3 is mostly nucleolar and has been directly implicated in cellular adaptation to mild oxidative stress[3,4]. More recently, SENP3 was also proposed as a negative regulator of autophagy during nutritional stress[5].

[1]Laboratory of Molecular Microbiology and Structural Biochemistry, Centre National de la Recherche Scientifique UMR5086, Université de Lyon, Lyon, France. [2]Laboratoire de Reproduction et Développement des Plantes, Université de Lyon, ENS de Lyon, UCBL, INRA, CNRS, Lyon, France. [3]Institut Pasteur, Université Paris Cité, Biologie des Bactéries Intracellulaires, CNRS UMR 6047, Paris, France. [4]Aix-Marseille Université, Inserm, Institut Paoli-Calmettes, CNRS, Centre de Recherche en Cancérologie de Marseille (CRCM), Marseille, France. [5]Institut Universitaire de France, Paris, France. [6]Laboratory of Molecular and Cellular Biochemistry, Meiji Pharmaceutical University, Tokyo, Japan. [7]Aix-Marseille Univ, CNRS, INSERM, CIML, Marseille, France. [8]Protein Science Facility, SFR Biosciences, Centre National de la Recherche Scientifique UAR3444, INSERM US8, Université de Lyon, ENS de Lyon, Lyon, France. [9]These authors contributed equally: Arthur Louche, Amandine Blanco. ✉e-mail: laurent.terradot@ibcp.fr; suzana.salcedo@ibcp.fr

Autophagy is one of the key stress-response mechanisms enabling cells to quickly adapt by selectively engulfing and degrading cellular components such as damaged organelles (e.g. mitophagy and ER-phagy), toxic protein aggregates (proteophagy), ribosomes (ribophagy), and microbes (xenophagy). Autophagy is a well-regulated process dependent on multiple protein complexes, notably on Beclin1, that acts as an essential mediator of autophagy initiation and nucleation steps and whose activity is tightly controlled directly or indirectly by post-translation modifications[6] as ubiquitination and SUMOylation[5].

Bacterial pathogens can escape autophagy-mediated killing and, in some cases, modulate autophagy components and pathways to promote virulence. One such example is *Brucella abortus*, an intracellular pathogen that extensively replicates inside cells in a vacuole derived from the endoplasmic reticulum (ER), known as replicative *Brucella*-containing vacuole (rBCV). Although the role of autophagy in *Brucella* virulence remains poorly understood, several proteins have been implicated in different stages of the intracellular life cycle, notably the formation of rBCVs (Atg9 and WIPI-1)[7] and induction of autophagic BCVs mediating bacterial egress from infected cells (ULK and Beclin1)[8]. The establishment of its replication niche requires the VirB type 4 secretion system (T4SS) that translocates effector proteins into host cells that specifically modulate cellular functions. Only very few *Brucella* effectors have been characterised to date, including some targeting innate immune responses[9,10] and the secretory pathway[11,12].

In this study, we identify two translocated effectors, NyxA and NyxB, that interact with SENP3, leading to its subnuclear mislocalisation during infection. We show that NyxA and NyxB mediate the formation of *Brucella*-induced foci (Bif) enriched in both effectors, the 60S ribosomal subunit protein RPL5 and the AAA-ATPase NVL. The formation of these structures is negatively regulated by SENP3, which *Brucella* sequesters to sustain intracellular replication.

The modulation of nuclear functions during infection is an important virulence strategy shared by a growing number of bacterial pathogens[13]. In most cases, this nuclear-targeting results in the fine regulation of host gene expression or control of the cell cycle to benefit host colonisation and persistence. In this study, we highlight a nucleomodulation mechanism that promotes perturbation of the subcellular localisation of nucleolar proteins during bacterial infection.

## Results

### The newly identified *B. abortus* effectors NyxA and NyxB accumulate in cytoplasmic and nuclear structures

Bacterial effectors often contain eukaryotic-like domains to enable efficient modulation of cellular pathways. Several bacterial effectors rely on a carboxyl-terminal CAAX tetrapeptide motif (C corresponds to cysteine, A to aliphatic amino acids and X to any amino acid) as a lipidation site to facilitate membrane attachment[14–17], with the presence of at least one carboxy-terminal cysteine being the essential feature and the remaining motif more flexible[18]. Previous work listed a subgroup of *Brucella* candidate effectors containing this kind of potential lipidation motif[16]. We have recently confirmed that one of these *Brucella* proteins, BspL, is translocated into host cells during infection[19]. Therefore, we set out to determine if two other *B. abortus* proteins, encoded by BAB1_0296 (BAB_RS17335) and BAB1_0466 (BAB_RS18145), could be translocated into host cells during infection. We relied on the TEM1 ß-lactamase reporter, widely used to assess the translocation of *Brucella* effectors in RAW macrophage-like cells, allowing high infection rates. At 24 h post-infection, we observed that BAB1_0296 was efficiently translocated into host cells, in contrast to BAB1_0466 (Fig. 1a, b). Hence, we can conclude that BAB1_0296 is likely to be a *B. abortus* effector, and we have named it NyxA, inspired by Greek mythology. Nyx is the Greek personification of the night, the daughter of Chaos, which seemed appropriate to us after remaining in the dark for so long regarding its function. It also alludes to its function as a *N*ucleomodulin autophag*y*-exploiting protein.

Genome analysis identified another gene, BAB1_1101 (BAB_RS21200), encoding for a protein with 82% identity to NyxA but without a carboxy-terminal CAAX motif (Supplementary Fig. 1a). For consistency, we have named it NyxB and found that it was also translocated by *B. abortus* into host cells, albeit to lower levels than NyxA at both 4 and 24 h post-infection (Fig. 1a, b and Supplementary Fig. 1b). NyxA and NyxB are well conserved within the *Brucella* species and show little homology to other bacterial proteins. Paralogs of NyxA and NyxB are present in members of the closely related genus *Ochrobactrum* from the *Brucellaceae* family.

To gain insight into the mechanism of translocation of NyxA and NyxB, we infected cells for 4 h, a time point that enables a comparison between the wild-type and the Δ*virB9* mutant strain lacking a functional T4SS. NyxA translocation was significantly reduced in a *virB* mutant strain, suggesting dependency on the T4SS. We should note that a small number of cells infected with the Δ*virB9*-expressing NyxA were positive for CCF2 compared to BAB1_0466, for which no translocation was detected, suggesting a proportion of NyxA could also be translocated independently of the T4SS (Supplementary Fig. 1b, c). Curiously, the translocation of NyxB was independent of the T4SS (Supplementary Fig. 1b).

To confirm that NyxA was indeed translocated across the vacuolar membrane during infection, we constructed new strains with NyxA fused on its N-terminus with a 4HA epitope tag, successfully used for imaging bacterial effectors[20]. We then infected HeLa cells, a well-characterised model of *B. abortus* infection nicely suited for microscopy studies. We could observe translocated 4HA-NyxA at 48 h post-infection, accumulating in cytoplasmic structures in the vicinity of multiplying bacteria (Fig. 1c). This was also the case for 4HA-NyxB (Fig. 1c, bottom panel). Analysis of fluorescence intensity profiles along a defined straight line across the 4HA-positive structures confirmed the majority of the 4HA signal detected does not correspond to intravacuolar NyxA or NyxB. For both effectors, punctate and filament-like structures were observed (Supplementary Fig. 1d) as early as 24 h post-infection. Equivalent NyxA structures were observed with an N-terminal 3Flag (Supplementary Fig. 1e), suggesting these are not an artefact due to the 4HA tag.

The 4HA-NyxA and NyxB positive structures could also be detected in the nucleus at 48 and 65 h post-infection (Fig. 1d), suggesting these effectors also target the host nuclei, particularly at the late stages of the infection.

### NyxA and NyxB target the same cellular compartments

Imaging of translocated NyxA and NyxB suggested that these effectors may have an identical subcellular localisation. To investigate this possibility, we ectopically expressed NyxA and NyxB with different tags. We found that a proportion of cells expressed HA-tagged NyxA and NyxB in cytosolic structures (Fig. 1e). Still, in most cells, HA-NyxA and NyxB predominantly accumulated in nuclear aggregates (Fig. 1e and Supplementary Fig. 2a), with cytoplasmic vesicular structures also visible. The co-transfection of HA-NyxA and myc-NyxB showed substantial co-localisation levels in the nucleus and cytosol (Fig. 1f and Supplementary Fig. 2b), suggesting that these two proteins target the same cellular compartments. In contrast, 4HA-tagged NyxA and NyxB were primarily found in the cytoplasm in vesicle-like structures (Supplementary Fig. 2a, c), with fewer cells showing nuclear aggregates (Supplementary Fig. 1b, c). This suggests that the presence of the 4HA reduces the nuclear import of NyxA and NyxB, revealing mostly its cytoplasmic location. This hypothesis was not confirmed in infected cells as effectors tagged with a single HA epitope were not detectable (Supplementary Fig. 2d).

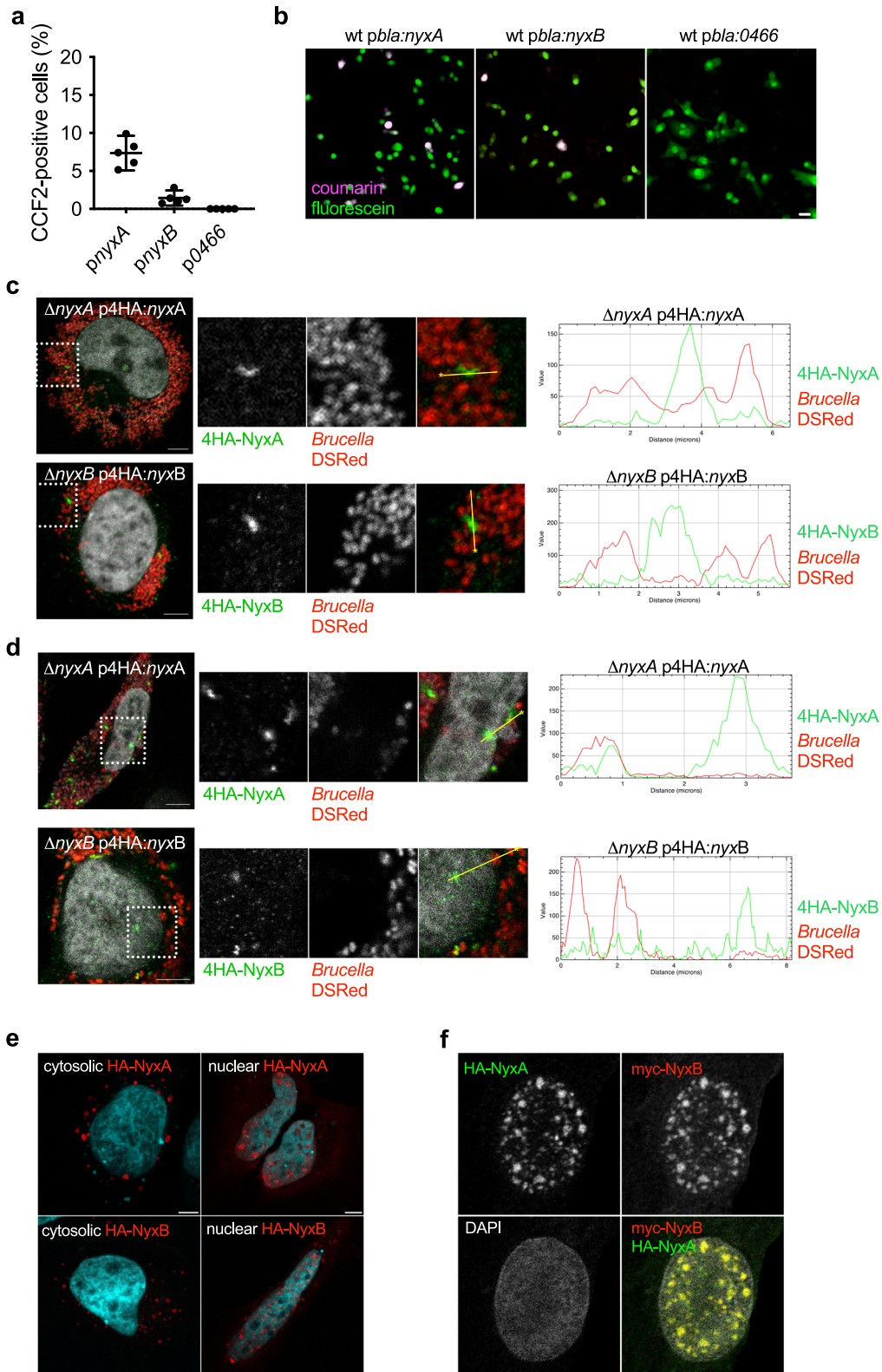

### The Nyx effectors interact with SENP3, which is necessary for efficient *B. abortus* intracellular multiplication

To investigate the potential role of NyxA and NyxB during infection, we constructed deletion mutants in *B. abortus* 2308. The overall intracellular bacterial counts of the single or double mutant strains were equivalent to the wild-type, suggesting that the lack of these effectors does not significantly impact the establishment of the replicative niche (Supplementary Fig. 3), often observed for

*Brucella* and other bacterial pathogens as *Legionella* with high effector redundancy. Therefore, to characterise the function of NyxA and NyxB, we identified their potential host-interacting partners by performing a yeast two-hybrid screen of NyxA against a library of human proteins (Supplementary Table 1). One of the main proteins identified was SENP3, and in view of its nuclear localisation and its diverse range of functions in key cellular processes and response to stress, we focused on this potential target.

**Fig. 1 | The *Brucella* NyxA and NyxB proteins are translocated into host cells during infection, accumulating in punctate or filament-like cytoplasmic and nuclear structures and can interact with each other. a** RAW macrophage-like cells were infected for 24 h with either *B. abortus* wild-type expressing TEM1 (encoded by the *bla* gene) fused with NyxA, NyxB or BAB1_0466. The percentage of cells with coumarin emission, which is indicative of translocation, was quantified after incubation with the CCF2-AM substrate. Data represent means ± 95% confidence intervals from five independent experiments, with more than 500 cells counted for each condition. **b** Representative images for *B. abortus* wild-type carrying p*bla:nyxA*, p*bla:nyxB* or p*bla:BAB1_0466* to exemplify the presence of effector translocation visible in coumarin-positive cells (violet) or absence of translocation. Scale bar corresponds to 25 μm. **c** Accumulation of 4HA-tagged NyxA (top) or 4HA-tagged NyxB (bottom) in cytoplasmic punctate or filament-like structures in HeLa cells infected for 48 h with Δ*nyxA* or Δ*nyxB* strains expressing DSRed and the

corresponding 4HA-tagged effector. The cell nucleus is visible with DAPI. A fluorescence intensity profile along a defined straight line across the 4HA-positive structures is included for each image, with the HA signal represented in green and the bacterial signal in red. **d** Representative confocal microscopy images showing accumulation of 4HA-tagged NyxA (top) or 4HA-tagged NyxB (bottom) in punctate nuclear structures in HeLa cells infected for 48 h with Δ*nyxA* or Δ*nyxB* strains expressing DSRed and the corresponding 4HA-tagged effector. **e** Representative confocal microscopy images of HA-tagged NyxA and NyxB (red) ectopically expressed in HeLa cells. Examples of predominant cytosolic localisation are on the left panels and nuclear localisation on the right panels. The nucleus of the cells is labelled with DAPI. Scale bars are 5 μm. **f** Confocal imaging showing co-localisation of HA-NyxA (green) and myc-NyxB (red) aggregates in the nucleus (white). All scale bars correspond to 5 μm.

SENP3 belongs to a family of cysteine proteases that share a conserved catalytic domain, characterised by a papain-like fold[21]. The variable N-terminal region often contributes to intracellular targeting of the protease. In the case of SENP3, the N-terminal region is implicated in nucleolar targeting as deletion of the residues 76–159 prevents nucleolar accumulation[22]. This region, rich in basic residues, contains the NPM1 binding domain and mTOR phosphorylation sites[22]. Two nucleolar localisation signals are predicted by the NoD server[23] at positions 23–46 and 106–137, the latter overlapping with the NPM1-binding site. The phosphorylation of SENP3 by the mTOR kinase facilitates interaction with NPM1, which enables subsequent nucleolar shuttling[22].

To confirm the interaction between SENP3 and the Nyx effectors, we attempted to purify SENP3. We were not successful and instead focused on the purification of the N-terminal region of SENP3 that encompassed all the yeast two-hybrid hits (SENP3$_{7-159}$). Both His-V5-tagged NyxA and NyxB were able to pull down SENP3$_{7-159}$, confirming a direct interaction of these effectors with the N-terminus domain of SENP3 (Fig. 2a). No unspecific binding to the column was detected (right panel, Fig. 2a). We cannot exclude the involvement of other regions of SENP3, but our data show SENP3$_{7-159}$ is sufficient for this interaction. To confirm these results, we carried out a co-immunoprecipitation from cellular extracts of HEK cells expressing myc-tagged NyxA or NyxB with either GFP or GFP-SENP3. NyxA and NyxB were efficiently co-immunoprecipitated with GFP-SENP3 and not GFP alone, confirming that NyxA/B and SENP3 are part of the same complex *in cellulo* (Fig. 2b).

As SENP3 seems to be the host target of both Nyx effectors, we determined its relevance during infection. The depletion of SENP3 was efficiently achieved after treatment with siRNA for 72 h (Fig. 2c), after which cells were infected with wild-type *B. abortus*. We observed a reduction in the percentage of cells with more than 10 bacteria at 48 h post-infection when SENP3 was depleted but not at earlier stages (Fig. 2d). A different set of siRNAs efficiently depleting SENP3 after 48 h (Fig. 2e) was also used to confirm these results (Fig. 2f). Therefore, in the late stages of the infection, SENP3 was required for *B. abortus* to multiply efficiently inside cells.

### The NyxB structure defines a novel family of effectors

To gain further insight into the function of NyxA and NyxB and their interaction with SENP3, we solved the crystal structure of NyxB at 2.5 Å (Fig. 3a, Supplementary Table 2). NyxA did not form crystals in any of the conditions tested. The asymmetric unit of the crystal contains 12 monomers of NyxB, but no significant differences were found between them and thus, only the structure of subunit A is described hereafter. The NyxB model encompasses residues 17–134, suggesting that residues 1–16 are flexible. NyxB has a mixed α-β fold with five β strands and six α-helices with a core made of helices α2–α4 and a small curved β-sheet formed by β3–β5. The longest helix α4 interacts with α2 and α6 and is connected to the core via a loop containing two short 3$_{10}$ helices

designated α3a and α3b (Fig. 3a). Helix α1 is loosely associated with the rest of the protein core and is positioned by the preceding and following loops. In particular, a β-hairpin formed by β1 and β2 packs against α5 and anchors α1 to the protein core. Search for structural homologues (DALI server and EMBL fold[24]) did not reveal any significant homology and thus makes of NyxB structure a prototype for this protein family. However, the C-terminal part of the protein showed some similarity with a number of nucleic acid-binding proteins, including Daschung, Ski or ribosomal protein RPL25 containing a winged helix domain. These similarities all match NyxB helices α2, α4 and α6 with the winged motif formed by strands β4 and β5.

Size-exclusion chromatography coupled to multi-angle light scattering (SEC-MALS) experiments indicate that both NyxA and NyxB form dimers (Supplementary Fig. 4A). Two putative dimers (dimers 1 and 2) were identified in the asymmetric unit (Supplementary Fig. 4b). The association of dimer 1 (chains A and H) buries a total of 530 Å$^2$ (Fig. 3b and Supplementary Fig. 4b), while dimer 2 (chains A and J) relies on fewer interactions burying a total of 400 Å$^2$ (Supplementary Fig. 4b). Small angle X-ray scattering on NyxB and NyxA proteins (Supplementary data, Supplementary Table 3 and Supplementary Fig. 4c, d) clearly showed that the two proteins adopt predominantly dimer 1 conformation in solution. This assembly relies on reciprocal hydrophobic and electrostatic interactions between α4 of one subunit and α6 of the other subunit and between the two α4–β4 loops (Fig. 3a). These results show that NyxA and NyxB are members of the same family and share a strong similarity in their tertiary and quaternary structures.

### Identification of the Nyx–SENP3-interacting residues

Taking advantage of the structural information of NyxB, we searched for potential interaction sites with SENP3. Analysis of the NyxB surface revealed an acidic pocket delineated by residues Y66, D80 and E82 within an acidic patch consisting of amino acids D72, E73, Y70, Y86 and Y63, residues that are strictly conserved in NyxA (Fig. 3c). In the context of the dimer, these surfaces are juxtaposed to form an extended concave negatively charged area of around 2000 Å$^2$ (Fig. 3d). To significantly modify the charge of the protein, we generated a triple mutant Y66R, D80R and E82R to obtain His-NyxB$^{MAG}$, where "MAG" stands for mutated acidic groove and Y62R, D76R and E78R to obtain His-NyxA$^{MAG}$. Purified NyxA or NyxB were able to pull down endogenous SENP3 from a HeLa cell extract, confirming their interactions (Fig. 3e). A decreased ability for both His-NyxA$^{MAG}$ and His-NyxB$^{MAG}$ to interact with SENP3 was observed (Fig. 3e). However, we could only detect a small amount of endogenous SENP3 in the cell extract with our antibody. As it is well established that when SENP3 is pulled down from a cell extract, its major cellular partner NPM1[25] can easily be detected by western blotting, we next probed the same membrane with an antibody against NPM1. NPM1 was indeed pulled down by His-NyxA and His-NyxB, but not when using His-NyxA$^{MAG}$ and His-NyxB$^{MAG}$, confirming that this mutation strongly impaired their ability to bind

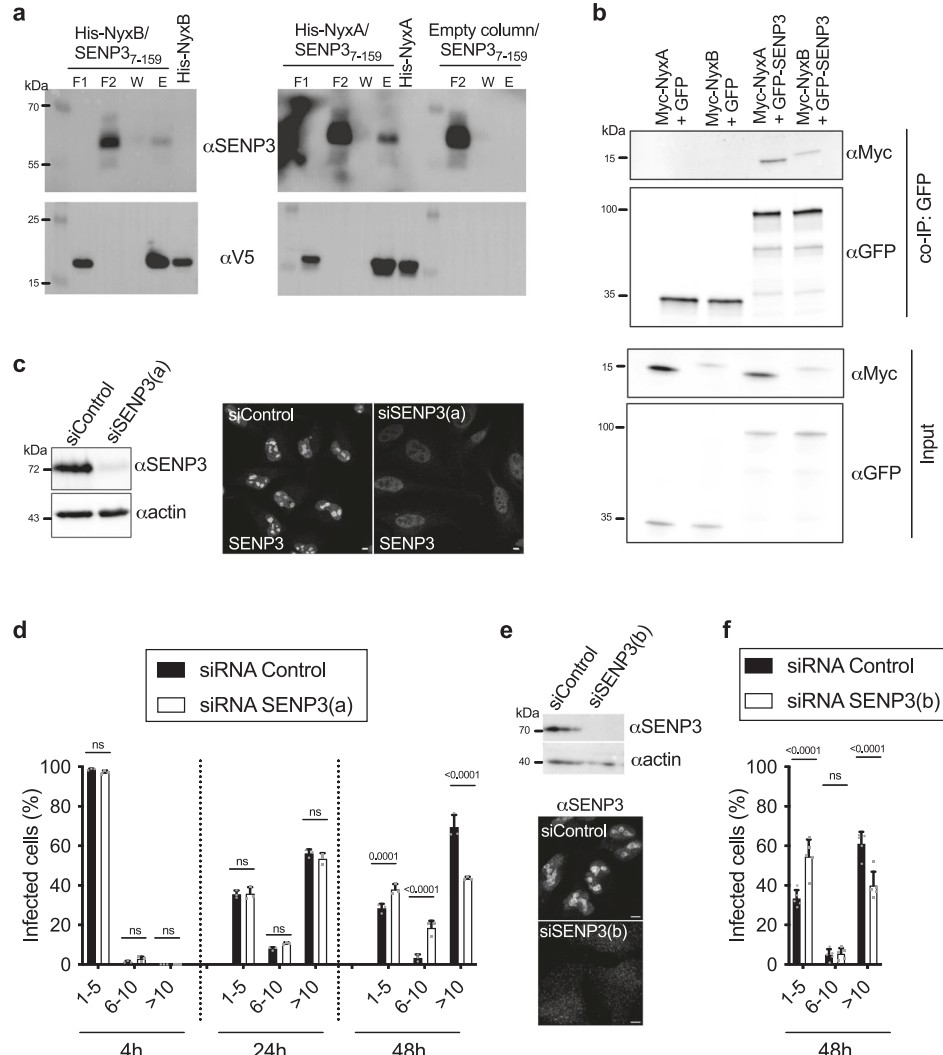

**Fig. 2 | The Nyx effectors interact with host protease SENP3, essential for efficient *B. abortus* intracellular multiplication. a** Pull-down assay with the N-terminal region of SENP3 from amino acid 7 until 159 (SENP3$_{7-159}$) against His-V5-NyxA or His-V5-NyxB immobilised on Ni NTA resins. An empty column was used as a control for non-specific binding and purified His-NyxA and His-Nyx-B inputs are shown. Interactions were visualised by western blotting using an anti-SENP3 antibody and column binding with anti-V5 (lower blot). Non-bound fractions (F1 and F2), last wash (W) and elution (E) are shown for each sample and the molecular weights indicated (kDa). **b** Co-immunoprecipitation (co-IP) assay from cells expressing GFP-SENP3 and Myc-NyxA or NyxB. GFP was used as a control for non-specific binding. The co-IP was revealed using an anti-Myc antibody, the fraction bound to GFP-trapping beads with an anti-GFP antibody, and the inputs (shown on the bottom two images) with anti-Myc and anti-GFP antibodies. Molecular weights are indicated (kDa). **c** Western blot of HeLa cell lysate treated with siRNA control (siControl) or siRNA SENP3 (siSENP3(a)) for 72 h. The membrane was probed with an anti-SENP3 antibody followed by anti-actin for loading control. Depletion was also verified by microscopy, showing a predominant nucleolar localisation of SENP3 in control cells which is strongly reduced in siSENP3(a) treated cells. Scale bar is 5 µm. **d** HeLa cells depleted for SENP3 or treated with the control siRNA for 72 h were infected with wild-type *B. abortus* expressing DSRed and the percentage of cells with either 1– 5, 6–10 or more than 10 bacteria per cell were quantified by microscopy at 2, 24 or 48 h post-infection. Data correspond to means ± 95% confidence intervals from three independent experiments, with more than 500 cells being counted for each siRNA treatment at each time point. A two-way ANOVA with Bonferroni correction was used to compare the bacterial counts obtained in siControl-treated cells with siSENP3 depleted cells, for each subgroup (1–5, 6–10 or >10 bacteria/cell) at each time-point. **e** Same as **c** but using an alternative siRNA mix (siSENP3(b)). Depletion was achieved after 48 h treatment. **f** Same as in **d** with the alternative siRNA mix treatment for 48 h and quantification of intracellular replication 48 h after. Data correspond to means ± 95% confidence intervals from five independent experiments. The *p* values are indicated in graphs (**d**) and (**f**), and not significant (ns) corresponds to *p* > 0.05.

the complex SENP3-NPM1 (Fig. 3e). As a negative control, the membrane was also probed for Histone 3, an abundant nuclear protein that did not bind to either NyxA or NyxB (Fig. 3e).

Together, these results confirm SENP3 as a target of the *Brucella* Nyx effectors and identify the acidic groove responsible for this interaction in vitro.

**The *Brucella* Nyx effectors induce delocalisation of SENP3**

After showing that NyxA and NyxB interact directly with SENP3, a eukaryotic protein mainly found in the nucleoli (Fig. 4a, top panel), we were intrigued by what impact these effectors could have on SENP3.

Ectopic expression of either HA-tagged NyxA or NyxB resulted in a marked reduction of endogenous nucleolar SENP3, which instead formed aggregates in the nucleoplasm (Fig. 4a, b). As SENP3 redistribution from nucleoli to the nucleoplasm could be due to starvation or mild oxidative stress[22,26], we ectopically expressed the mutant HA-NyxA$^{MAG}$, unable to interact with SENP3. The mutation of the acidic interaction groove impaired the delocalisation of SENP3 by NyxA and, to a lesser extent, NyxB, confirming this effect was due to direct interaction with SENP3. Consistently, analysis of the same images for co-localisation of SENP3 with HA-tagged Nyx proteins revealed important recruitment, dependent on the acidic groove (Fig. 4c).

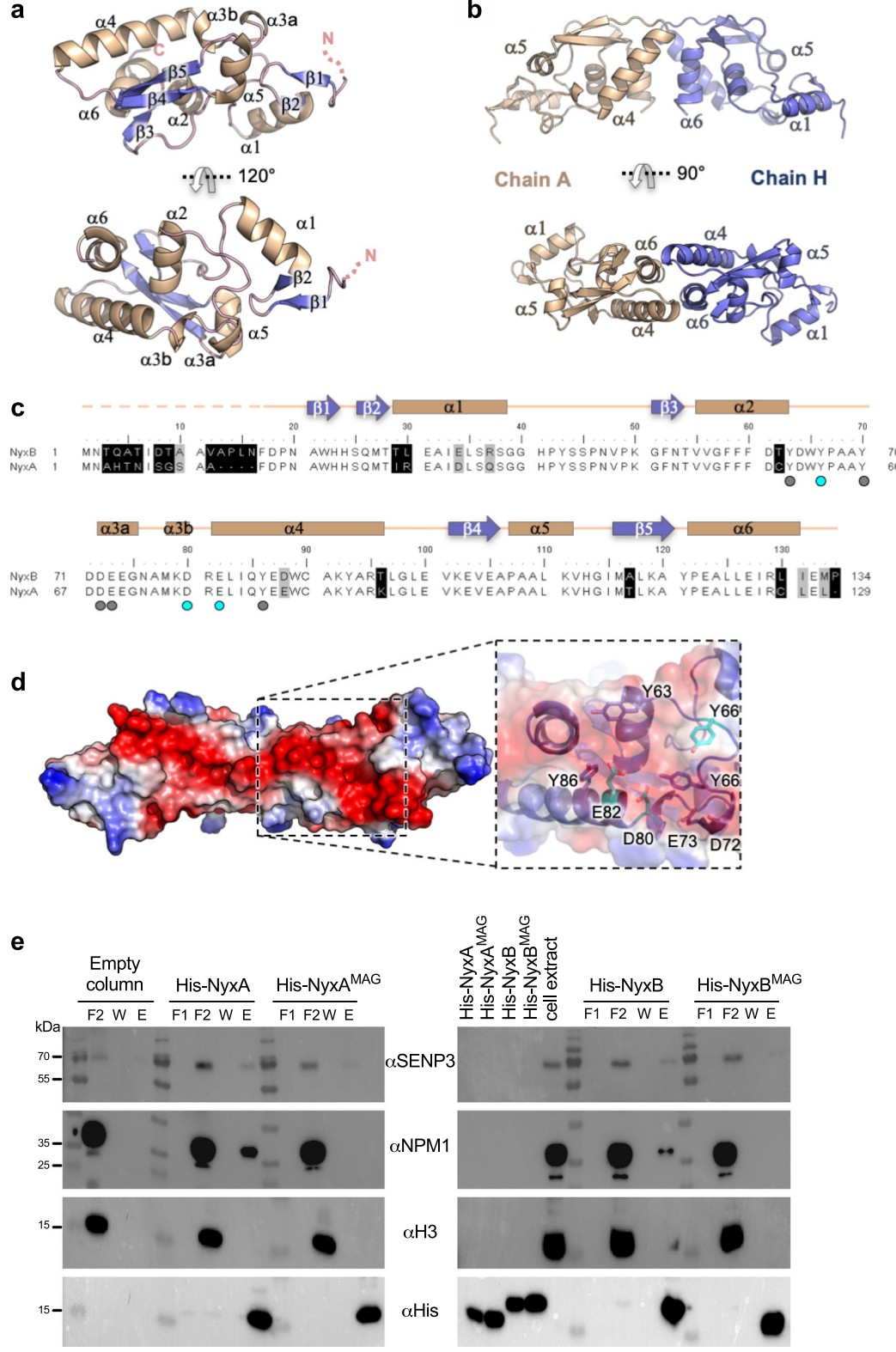

Next, we investigated the prevalence of these phenotypes during infection. We infected HeLa cells with either the wild-type *B. abortus* strain, the mutant lacking *nyxA*, or a complemented strain expressing *nyxA* from the chromosome under the control of its promoter. Analysis of the level of SENP3 retained in the nucleoli during infection showed a significant lack of nucleolar localisation of SENP3 in cells infected with the wild-type *B. abortus* strain in contrast with the Δ*nyxA* strain (Fig. 4d, e). The wild-type phenotype was partially restored with

the complemented strain but not with a Δ*nyxA* strain expressing *nyxA*MAG. The absence of complementation observed for the strains expressing NyxAMAG was not due to a lack of translocation, as TEM-NyxAMAG and TEM-NyxBMAG were both efficiently translocated during infection (Supplementary Fig. 5a). This result shows that NyxA interaction with SENP3 during infection prevents its accumulation in nucleoli. In the case of NyxB, we could also observe a statistically significant increase of SENP3 in the nucleoli in cells infected with the

**Fig. 3 | The NyxB structure defines a novel family of effectors allowing the identification of the SENP3 interacting groove. a** Two views of the NyxB monomer depicted in ribbon with helices coloured in wheat, strands in blue and loops in pink. **b** Two views of the NyxB dimer. **c** Structure-based sequence alignment of NyxB and NyxA. Secondary structure elements are indicated above the sequences. Identical residues are not shaded, residues shaded in black and grey is non-conserved and conserved, respectively. Dots indicate residues identified in the acidic patch and cyan dots indicate acidic groove mutants (MAG). **d** Surface representation of NyxB dimer coloured according to electrostatic potential (red negative, blue positive) showing the extended acidic patch. The inset shows a close-up view of the area with residues' side chains displayed as ball-and-sticks and mutated residues (E82, Y66 and D80) coloured in cyan. **e** Pull-down assay with His-NyxA, His-NyxB or the specific catalytic mutants (His-NyxA$^{MAG}$ or His-NyxB$^{MAG}$) immobilised on Ni NTA resins that were incubated with a HeLa cell extract. An empty column was used as a control for non-specific binding. Interactions with endogenous SENP3, NPM1 or Histone 3 (H3) were visualised by western blotting using the corresponding antibody and column binding with anti-His (lower blot). Non-bound fractions (F1 and F2), last wash (W) and elution (E) are shown for each sample and the molecular weights indicated (kDa). The cell extract and the different purified Nyx inputs are also shown.

$\Delta nyxB$ strain compared to cells infected with wild-type *B. abortus*, although to a lesser extent than what we observed for $\Delta nyxA$. However, we could not fully complement this phenotype (Supplementary Fig. 5b), possibly due to the low sensitivity of this microscopy approach combined with a weaker phenotype. As expected, a strain lacking both genes encoding for NyxA and NyxB could not mislocalise SENP3 as the wild-type strain (Supplementary Fig. 5b). Representative images of all strains are shown in Supplementary Fig. 5c.

### Nyx effectors induce cytoplasmic accumulation of the nucleolar protein NVL in *Brucella*-induced foci (Bif)

One of the principal roles of SENP3 in the nucleoli is to regulate ribosomal biogenesis, specifically of the 60S ribosomal subunit[27]. Briefly, mammalian 80S ribosomes result from the assembly of a large 60S subunit, composed of 5S, 5.8S and 28S rRNAs, and a small 40S subunit comprised of the 18S rRNA. A high number of ribosomal proteins are associated with each subunit. SENP3 is implicated in promoting the maturation of the 28S rRNA by de-SUMOylating several nuclear proteins, including NPM1[25]. The observation is that SENP3 was unable to accumulate in the nucleoli during *B. abortus* infection prompted us to investigate if NyxA and NyxB could impact other nucleoli proteins associated with the biogenesis of the 60S ribosomal subunit, such as SENP3. We did not observe any effect on the nucleolar levels of Pescadillo (PES1) (Supplementary Fig. 6a), involved in the maturation of the 28S and 5.8S rRNAs and that interacts with NPM1 for nucleolar targeting but does not interact with SENP3[28]. This suggests that not all nucleolar ribosomal biogenesis-associated proteins are impacted during *Brucella* infection. A slight effect on NPM1 nucleolar accumulation was observed in some *B. abortus* infected cells (Supplementary Fig. 6b), but to a much lower extent than SENP3, suggesting that the bulk of nucleolar NPM1 remained unaffected.

We next analysed the nucleolar accumulation of the VCP-like AAA-ATPase (NVL), also part of the complex of proteins involved in the maturation of the 28S and 5.8S rRNAs[29]. Although significant NVL staining was still detected in the nucleoli of infected cells, we observed a striking cytoplasmic accumulation of NVL in punctate structures in all *B. abortus* infected cells at 48 h post-infection that were rarely visible in non-infected cells (Fig. 5a). We have named these structures *Brucella*-induced foci (Bif).

To determine if the *Brucella* Nyx effectors contributed to the formation of NVL-positive Bif, we quantified their number in HeLa cells infected either wild-type or a mutant lacking both NyxA and NyxB. We found that cells infected with $\Delta nyxA nyxB$ had fewer NVL-positive Bif (Fig. 5b), suggesting the Nyx effectors contribute to their induction. Using single Nyx mutant strains to assess the contribution of each effector, we found that both effectors contribute to the formation of NVL-positive Bif (Fig. 5b). These differences were not due to different intracellular bacterial numbers as *nyx* mutant strains replicate as efficiently as the wild-type in HeLa cells (Supplementary Fig. 3). Importantly, both mutants were fully complemented (Fig. 5b). However, expression of either NyxA and NyxB carrying the MAG mutations that are unable to interact with SENP3, failed to complement the mutant phenotypes (Fig. 5b).

To confirm this phenotype, we infected immortalised bone marrow-derived macrophages (iBMDMs), a well-established model of *Brucella* infection[19,30]. Strong induction of NVL cytoplasmic punctate accumulation was observed in iBMDMs infected with wild-type *B. abortus* (Fig. 5c) in a Nyx-dependent manner (Fig. 5d).

### *Brucella*-induced NVL foci are also enriched in the ribosomal protein RPL5 and the Nyx effectors

NVL interacts with the ribosomal protein 5 (RPL5), a component of the 60S ribosomal subunit responsible for transporting NVL to the nucleoli[31]. To test if RPL5 could be retained in the cytoplasm, we analysed RPL5 distribution in cells infected with wild-type *B. abortus*. We observed the same striking accumulation of RPL5 in cytoplasmic structures in all infected HeLa cells (Supplementary Fig. 7a) and bone marrow-derived macrophages (Supplementary Fig. 7b). Indeed, RPL5-positive Bif were present in even higher numbers than observed for NVL (Supplementary Fig. 7c). The formation of these cytoplasmic structures was also dependent on the Nyx effectors as a strain lacking both NyxA and NyxB showed a reduced number of RPL5-positive foci at 48 h post-infection (Supplementary Fig. 7c). As these structures were reminiscent of the localisation of translocated 4HA-NyxA and NyxB (Fig. 1c), we infected cells with strains expressing 4HA-tagged effectors and co-stained for NVL or RPL5. Indeed, NVL significantly co-localised with the 4HA-tagged Nyx effectors in both HeLa (Fig. 5e) and macrophages (Supplementary Fig. 8), as well as when using 3Flag-tagged NyxA (Supplementary Fig. 9). Similarly, RPL5-positive cytoplasmic structures induced upon *Brucella* infection contained NyxA and NyxB (Supplementary Fig. 7d). Due to antibody incompatibility, we could not co-label NVL and RPL5 simultaneously. Nonetheless, these data overwhelmingly support that Bif are enriched in NVL, RPL5, and both Nyx effectors during infection. However, the presence of NyxA/B alone is insufficient to induce this phenomenon as ectopically expressed NyxA and NyxB have no impact on NVL and RPL5 distribution, even when expressed simultaneously (Supplementary Fig. 9c).

To determine if NyxA/B could interact with NVL, we conducted a pull-down experiment with the purified effector proteins. Unlike what was observed for SENP3, NVL was not found in the eluted samples suggesting it is not part of the Nyx–SENP3 complex (Supplementary Fig. 10a). However, the use of purified proteins and total cell extracts for in vitro pull-down experiments may prevent the detection of complexes being transiently formed *in cellulo* or requiring specific cellular co-factors. Therefore, we took advantage of the proximity ligation assay (PLA), a method used to assess protein–protein interactions in situ[32]. As the Nyx cytosolic structures can only be detected during *Brucella* infection, we carried out the PLA in cells infected with *B. abortus* expressing 3Flag-NyxA, at 48 h post-infection. These experiments were undertaken with the mouse anti-FLAG antibody (Sigma, M2) and rabbit anti-NVL and RPL5 antibodies[31,33] as we validated their specificity in immunofluorescence microscopy (Supplementary Fig. 11). Despite non-specific small bright dots in all controls (Supplementary Fig. 12a), a clear PLA-positive signal was observed for

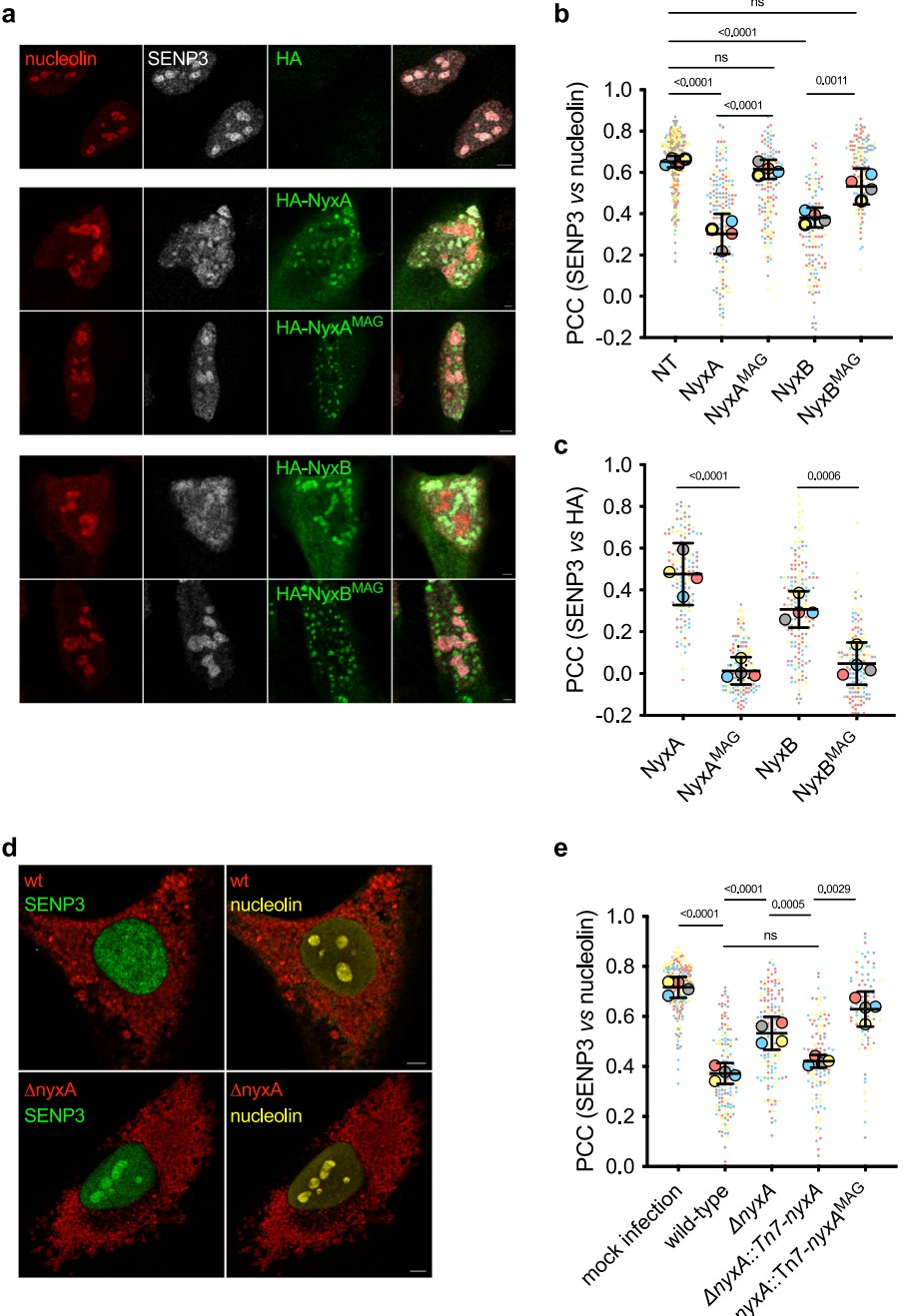

**Fig. 4 | The *Brucella* Nyx effectors directly perturb the SENP3 nucleolar localisation in host cells, including during infection. a** Representative confocal microscopy images of HeLa cells expressing the HA empty vector, HA-NyxA, HA-NyxA$^{MAG}$, HA-NyxB and HA-NyxB$^{MAG}$. Nucleolin (red), SENP3 (white) and HA (green) were revealed with specific antibodies. **b** Quantification of the Pearson correlation coefficient of SENP3 versus nucleolin (see methods for plugin description). Data are represented as means ± 95% confidence intervals from four independent experiments. Each experiment is colour coded and all events counted are shown. Data were analysed using one-way ANOVA by including all comparisons with Tukey's correction. Not all comparisons are shown. All the cells quantified are shown in the format of SuperPlots, with each colour representing an independent experiment and its corresponding mean ($N = 4$). **c** The same data set as in **b** was used for quantification of the Pearson correlation coefficient of SENP3 versus HA to assess recruitment and data represented and statistical comparison are as described in (**b**). **d** Representative confocal microscopy images of HeLa cells infected for 48 h with wild-type DSRed expressing *B. abortus* or Δ*nyxA*, with nucleolin (yellow) and SENP3 (green). **e** Quantification of the Pearson's coefficient of SENP3 versus nucleolin in HeLa cells infected for 48 h with either *B. abortus* wild-type or Δ*nyxA*, its complemented strain Δ*nyxA*::Tn7-*nyxA* or a complementing strain expressing the mutated acidic groove responsible for interaction with SENP3 (Δ*nyxA*::Tn7-*nyxA*$^{MAG}$). Data are represented and analysed as in (**b**). Not all comparisons are shown. All microscopy images displayed have scale bars corresponding to 5 μm. The p values are indicated in graphs **b**, **c** and **e**, and not significant (ns) correspond to $p > 0.05$.

both NVL and RPL5 (Fig. 5f), confirming these are within 40 nm of 3FLAG-NyxA and likely to be part of the same complex. No such signal was detectable in the controls (Supplementary Fig. 12a). The NyxA-NVL/RPL5 PLA-positive structures were more heterogeneous than

those observed using standard immunofluorescence labelling, probably due to the signal amplification obtained with the PLA (Fig. 5f). We could also detect labelling surrounding the bacteria, suggesting an association with the *Brucella*-containing vacuole membrane (Fig. 5f,

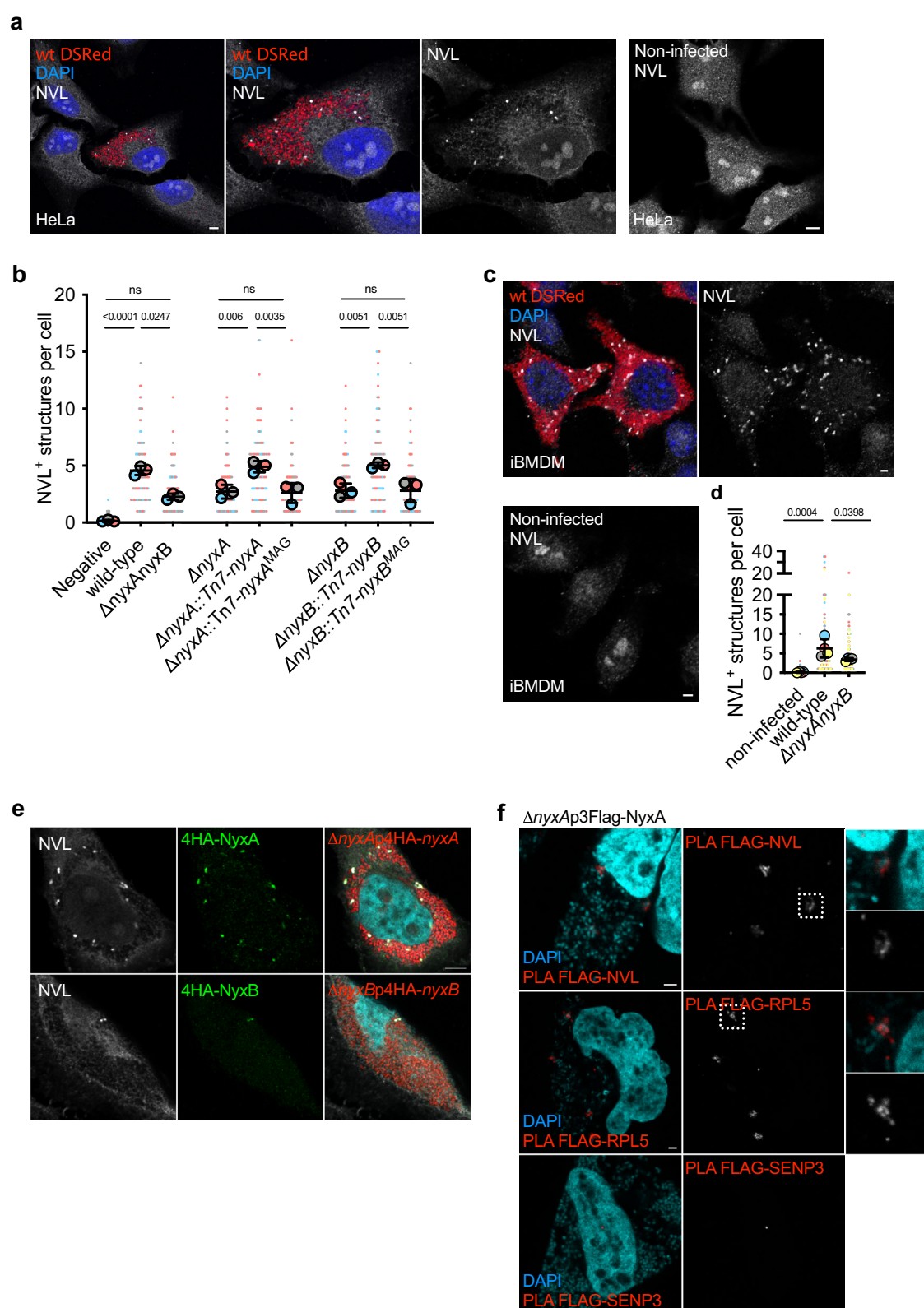

zoom inlets). In contrast, no PLA signal was detected with SENP3, either using our normal labelling protocol (same as for NVL and RPL5, Fig. 5f) or a protocol to specifically label cytosolic SENP3[5] (Supplementary Fig. 12b), which we confirmed as efficient (Supplementary Fig. 13). These results suggest that the cytosolic structures induced during *Brucella* infection are enriched in Nyx, NVL and RPL5 but not SENP3, which remains mostly confined to the nuclei.

## NyxA/B-mediated induction of Bif and de-localisation of nucleolar proteins is dependent on Beclin1 and negatively regulated by SENP3

Formation of cytoplasmic aggregates has been described in several bacterial infections, including stress granules[34], P-bodies[35] and U-bodies[36]. None of these was observed at 48 h post-infection (Supplementary Fig. 14a, b) when Bif are normally present. Labelling with

**Fig. 5 | *B. abortus* NyxA and NyxB induce cytoplasmic punctate accumulation of NVL and RPL5 in *Brucella*-induced foci (Bif). a** Confocal microscopy image of wild-type *B. abortus* (expressing DSRed) infected HeLa cells in comparison with non-infected cells or **c** immortalised bone-marrow derived macrophages (iBMDM) labelled with an anti-NVL antibody (white). **b** Quantification of the number of NVL-positive cytoplasmic structures in mock-infected control HeLa cells in comparison to wild-type or a mutant strain lacking each or both *nyxA* and *nyxB*, complemented strains or expressing MAG mutants. Data are represented as means ± 95% confidence intervals from three independent experiments. Each experiment is colour coded and all events counted are shown. Data were analysed using one-way ANOVA by including all comparisons with Tukey's correction. Not all comparisons are

shown but are available in Supplementary Table 4. **d** Same as in **b** in iBMDMs, focusing on control mock-infected cells in comparison to iBMDMs infected with wild-type or a mutant lacking NyxA/B. **e** Representative confocal microscopy images of HeLa cells infected with either Δ*nyxA* expressing DSRed and 4HA-NyxA (top) or Δ*nyxB* expressing DSRed and 4HA-NyxB for 48 h and labelled for NVL (white) and DAPI (cyan). **f** Proximity ligation assay (PLA) in HeLa cells infected for 48 h with Δ*nyxA* expressing 3Flag-NyxA, with either NVL (antibody from M. Nagahama, top panel), RPL5 (antibody from M. Nagahama, middle panel) or SENP3 (bottom panel). Zoom inlets correspond to positive PLA signal observed surrounding individual bacteria. All scale bars correspond to 5 μm. The *p* values are indicated in graphs (**b**) and (**d**), and not significant (ns) corresponds to *p* > 0.05.

the FK2 antibody revealed the presence of aggregates of mono- and poly-ubiquitinated proteins in some *B. abortus* infected cells, but these did not co-localise with NVL (Supplementary Fig. 14c). We next focused on other cellular processes known to result in cytoplasmic accumulation of nucleolar proteins. Recent reports have described NUFIP1 as a nucleo-cytoplasmic shuttling protein that accumulates in the cytoplasm upon starvation. In this context, NUFIP1 acts as a receptor for ribophagy, a specialised autophagy process dedicated to the degradation of ribosomes to generate nutrients[37]. Given the presence of RPL5 in these structures, a component of the 60S ribosomal subunit, we hypothesised that *B. abortus* infection could result in induced ribophagy. We first reported the increase in cytoplasmic NUFIP1 in cells infected with wild-type *B. abortus* compared to non-infected cells, with the appearance of bright punctate structures dependent on NyxA and NyxB (BioRXiv: https://doi.org/10.1101/2021.04.23.441069). However, analysis of labelling in HEK NUFIP knockout cells during the revision of this manuscript releveled a very high level of non-specific immunofluorescence labelling (Supplementary Fig. 15) with the commercial antibody we had used and that was previously used by others[38]. This has prevented us from concluding if NUFIP is present in Bif, and the data has been removed from the current manuscript. In addition, a recent study has questioned the role of NUFIP as a ribophagy receptor[39].

We next assessed if these structures could result from enhanced autophagy. Using siRNA we managed to partially deplete the autophagy-initiation protein Beclin1 (Supplementary Fig. 16), which is essential for initiating autophagy. We quantified the number of Bif in wild-type infected cells at 48 h post-infection, using NVL as a marker. We found a clear reduction in the number of NVL-positive Bif in the absence of Beclin1 (Fig. 6a), consistent with an autophagy-derived process. To determine if Bif could correspond to autophagosomes, we labelled with either lysosomal-associated membrane protein 1 (LAMP1) to detect lysosomes or the microtubule-associated protein light chain 3 (LC3), a commonly used marker for autophagosomes. No co-localisation of LAMP1 was observed with either NVL or RPL5 at 48 h post-infection nor with 4HA or 3FLAG-tagged NyxA (Supplementary Fig. 17), suggesting the absence of autophagosome formation. Consistently, we did not observe any significant co-localisation between translocated 3FLAG-NyxA and LC3 (Supplementary Fig. 18a) nor with NVL or RPL5 at 48 h post-infection, using cells expressing LC3-GFP to avoid methanol fixation (Supplementary Fig. 18b). However, we observed a Bif juxtaposition with large LC3-positive vacuoles (arrows, Supplementary Fig. 18a). Together these data suggest Bif are not degradative autophagosomes. The absence of ER and mitotracker in most Bif indicates that these structures are not directly derived from ER or Mitophagy processes (Supplementary Fig. 19).

Recently, SENP3 has been shown to de-SUMOylate Beclin1 in the cytoplasm upon starvation and negatively regulate its activity, acting as a switch-off mechanism to prevent excessive autophagy[5]. NyxA and NyxB retention of SENP3 preventing its nucleolar accumulation are also likely to lead to decreased activity of SENP3 in the host cytoplasm, which could impact its ability to regulate Beclin1. Therefore, to determine if the absence of SENP3 activity could account for the

induction of NVL cytoplasmic structures, we depleted SENP3 and infected HeLa cells with wild-type *B. abortus*. The depletion of SENP3 resulted in a substantial rise in cytoplasmic NVL-positive Bif being formed as early as 24 h post-infection (Fig. 6b). At 48 h post-infection, wild-type infected cells showed a very high number of Bif in the absence of SENP3, with some cells containing more than 100 puncta (Fig. 6c, d). Importantly, depletion of SENP3 alone was insufficient to induce NVL cytoplasmic accumulation, suggesting this phenotype is triggered by the infection.

We could not determine if Beclin1 is present in Bif due to a lack of specific anti-Beclin antibody labelling compatible with the immunofluorescence detection of these structures. However, Beclin1 is predominantly SUMO-conjugated by the E3 ligase PIAS3 in conditions of nutritional stress regulated by SENP3[5]. We, therefore, depleted cells for PIAS3 (Supplementary Fig. 516b). A significant reduction in the formation of NVL-positive structures was observed at 24 h post-infection, equivalent to the effect of depleting Beclin1 (Fig. 6e). This phenotype contrasts with the depletion of SENP3 using an alternative siRNA, which strongly induces the formation of NVL-positive Bif (Fig. 6e). Although our attempts to establish the SUMOylation status of Beclin1 in infected cells were not successful, probably due to low infection rates of *Brucella*, we found that intracellular replication at 48 h post-infection was reduced in the absence of either Beclin1 or PIAS3 suggesting their functions contribute to efficient multiplication (Fig. 6f).

Given our results, we hypothesised that cytosolic accumulation of NVL could constitute a generalised cellular response to intracellular bacteria. We selected another intracellular pathogen, *Legionella pneumophila* that multiplies in an ER-derived vacuole as observed for *B. abortus*[40]. However, infection with *L. pneumophila* did not result in NVL cytoplasmic accumulation (Supplementary Fig. 20), suggesting this phenomenon is specifically induced during *B. abortus* infection.

In summary, these data show that Nyx-mediated accumulation of nucleolar proteins in cytoplasmic structures induced during *Brucella* infection is dependent on Beclin1 and PIAS3, is negatively regulated by SENP3 and contributes to efficient intracellular multiplication of *B. abortus*.

## Discussion

Here we report the identification of two *Brucella* effectors that target SENP3 and induce its delocalisation from the nucleoli during infection. The action of NyxA and NyxB enhances the appearance of cytoplasmic structures enriched in NVL and RPL5, whose formation is dependent on Beclin1 and PIAS3 and negatively regulated by SENP3. We, therefore, propose that NyxA/B interaction with SENP3 in the nucleus and its subsequent retention in the nucleoplasm promotes the formation of these *Brucella*-induced cytosolic structures (Fig. 6g). Our data suggest that SENP3 contributes to the *Brucella* intracellular lifecycle.

How these effectors are translocated into host cells remains unclear. The TEM-fusion assays indicate that the translocation of NyxA is mostly dependent on the T4SS, unlike NyxB. Given the high sequence identity between the two proteins and their structural similarity in SAXS analysis, it is likely that the two proteins adopt very

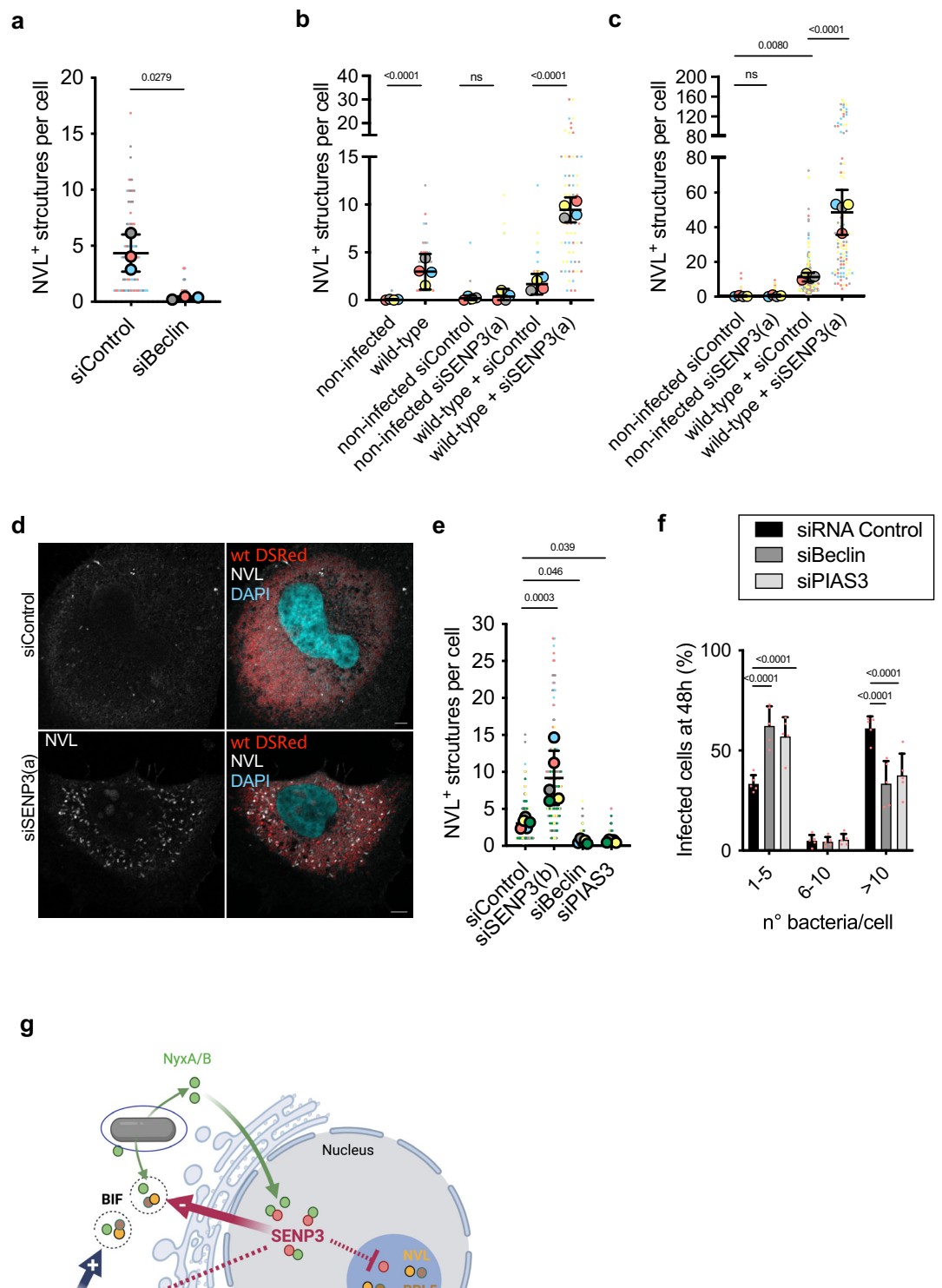

similar ternary and tertiary structures and thus the main differences between these two effectors reside in the N-terminal domain and the last 5 C-terminal residues. A T4SS secretion signal may be present within these regions. Unfortunately, we could not use the 4HA-tagged Nyx effectors to dissect their secretion mechanisms as they were not detected at the early stages of the infection when the numbers of wild-type and *virB* mutant bacteria are equivalent. We are currently working on alternative reporter systems with higher sensitivity to better

understand the kinetics and mechanisms of translocation, as NyxA and NyxB may constitute an interesting tool to understand the molecular basis of substrate selection by the *Brucella* VirB T4SS.

The discovery of NyxA was based on the presence of a potential CAAX motif. However, NyxB, which has the same intracellular location and function, and is more than 80% identical in amino acid sequence, does not contain this putative lipidation site. Therefore, we have no evidence to date that this sequence is important for NyxA function, nor

**Fig. 6 | Formation of Bif is dependent on Beclin1 and negatively regulated by SENP3. a** Quantification of the number of NVL-positive structures in cells infected by the wild-type *B. abortus* and depleted for Beclin1 with siRNA or treated with scrambled siRNA (siControl). Data are represented as means ± 95% confidence intervals from three independent experiments. Data were analysed using one-way ANOVA by including all comparisons with Tukey's correction. **b** Quantification of the number of NVL-positive cytoplasmic structures in HeLa cells pre-treated with control siRNA or siSENP3(a) and infected for an additional 24 h with wild-type *B. abortus* or **c** 48 h. Mock-infected cells are included as controls. Data are represented as means ± 95% confidence intervals from four independent experiments. Each experiment is colour coded and all events counted are shown. Data were analysed using one-way ANOVA by including all comparisons with Tukey's correction. Not all comparisons are shown. **d** Representative images of NVL cytoplasmic punctate accumulation (white) in HeLa cells treated with control siRNA (top) or depleted for SENP3 (bottom) followed by infection for 48 h with *B. abortus* expressing DSRed. Scale bars correspond to 5 μm. **e** Quantification of the number of NVL-positive Bif in HeLa cells pre-treated with control siRNA or an alternative siSENP3(b) mix, siBeclin or siPIAS3 and infected for an additional 24 h with wild-type *B. abortus*. Data correspond to means ± 95% confidence intervals from five independent experiments.

Data were analysed using one-way ANOVA by including all comparisons with Tukey's correction. **f** HeLa cells depleted for Beclin or PIAS3 or treated with the control siRNA for 48 h were infected with wild-type *B. abortus* expressing DSRed and the percentages of cells with either 1–5, 6–10 or more than 10 bacteria per cell were quantified by microscopy at 48 h post-infection. Data correspond to means ± 95% confidence intervals from five independent experiments. A two-way ANOVA with Bonferroni correction was used to compare each condition to the siControl-treated cells. The *p* values are indicated in the graphs (**a**–**f**), and not significant (ns) corresponds to *p* > 0.05. **g** NyxA/B are translocated into host cells, being detected on the vacuolar membrane, in the vicinity of replicating bacteria and in the nucleus. Within the nucleus, NyxA/B interact with SENP3, retaining it in the nucleoplasm and preventing its accumulation in the nucleoli. This sequestration promotes the formation of Bif, which are novel cytosolic structures induced upon *Brucella* infection, enriched in the Nyx effectors and the ribosomal proteins RPL5 and NVL. SENP3 acts as a negative regulator of Bif whereas Beclin and PIAS3 activities are required for Bif formation. The effect of NyxA/B sequestration of SENP3 on the SUMOylation levels of Beclin remains to be determined. Created with Biorender.com.

---

if it constitutes a functional lipidation motif. NyxA does contain the critical conserved cysteine within this motif, followed by the aliphatic amino acid leucine. However, one glutamic acid is also present. Preliminary experiments from our lab show no difference in intracellular localisation of NyxA lacking the C-terminal last four amino acids, consistent with the possibility that it is unnecessary for membrane targeting. The translocated Nyx effectors were not significantly enriched on the vacuole membrane, enclosing replicating bacteria or the plasma membrane. Instead, 4HA-NyxA/B is localised to punctate cytoplasmic structures and the nucleus. Interestingly, cytoplasmic foci were detectable at 24 h after infection, whereas nuclear aggregates were only visible at 48 h and onwards, suggesting that cytoplasmic foci precede the nuclear import. However, our transfection data suggest that the presence of the 4HA tag significantly reduces nuclear import. So, it is likely that native untagged NyxA/B are efficiently targeting both nuclear and cytoplasmic compartments during infection.

NyxA/B are two *Brucella* nucleomodulins impacting the subnuclear localisation of SENP3 and subsequently of other nucleolar proteins such as RPL5 and NVL. Although both effectors contribute to these phenotypes, deletion of both genes is insufficient to obtain the phenotype of non-infected cells. These results suggest that additional effectors may be involved in SENP3 delocalisation, either directly or indirectly, which would explain why a mutant lacking both NyxA and NyxB does not have a replication defect. On the other hand, depletion of SENP3 with two distinct siRNA results in a very high accumulation of Bif and renders *B. abortus* intracellular replication less efficient. One possible explanation is that the accumulation of a too large amount of Bif is harmful to the cell and/or the bacteria. Alternatively, prolonged siRNA depletion of SENP3 from the early stages of the infection to 48 h post-infection indirectly impacts *Brucella* replication, for example, by decreasing ribosomal biogenesis and host protein synthesis[25]. This depletion would not correspond to a timed effect of translocated NyxA/B, which does not deplete SENP3 from infected cells to the levels of the siRNA treatments.

Although ectopic expression of the Nyx effectors alone is sufficient to delocalise SENP3, it is not enough to induce Bif formation. *Brucella* infection must cause stress or activate a specific signalling pathway that, in the context of the NyxA/B interaction with SENP3, results in the formation of Bif in infected cells. Although *Brucella* is considered a stealthy pathogen, its intracellular trafficking and replication generate several stresses in the cell, including ER stress[41]. Extensive intracellular multiplication of *Brucella* is likely to lead to starvation-like conditions. Indeed, important metabolic switches have been described during infection[42,43]. Infected cells have been shown to have a reduced ability to utilise amino acids suggesting intracellular *B.*

*abortus* is competing with the cell for nutrients[44], and additional studies support the notion that *Brucella* is using amino acids from the host[44,45].

A recent study has implicated SENP3 in the regulation of autophagy[5]. During starvation conditions, SENP3 is delocalised to the host cytoplasm, where it de-SUMOylates Beclin1, which reduces its interaction with other autophagy components, acting as a negative regulator of autophagy. *Brucella* extensive replication may induce significant stress reminiscent of starvation conditions that would, in turn, trigger an autophagy-derived process. The nuclear retention of SENP3 by NyxA and NyxB could prevent this negative regulator from reaching the cytoplasm, de-SUMOylating Beclin1, and switching off autophagy during *Brucella* infection. This is consistent with the requirement of the SUMO E3 ligase PIAS3 for the formation of Bif, which SUMOylates Beclin during starvation, a process negatively regulated by SENP3[5]. However, further work is required to establish the impact of SUMOylation in *Brucella* pathogenesis and assess the SUMOylation status of Beclin during infection. We are yet to identify the specific targets of SENP3 that would be directly affected by its interaction with NyxA/B.

The presence of RPL5 in Bif and the requirement of Beclin1 for their formation led us to originally hypothesise that these could be ribophagy-derived. Induction of ribophagy during *Brucella* infection could provide a source of nutrients to the cell and potentially replicating bacteria. Curiously, proteomics studies have shown a reduction in bacterial ribosomal proteins during the early stages of *Brucella* infection, which could indicate *Brucella* is using them as a nutrient source[44]. *Brucella* could be degrading the cell's ribosomes at later stages of the infection cycle to sustain massive intracellular replication, but this remains speculative. Additional effectors could have enzymatic activities or recruit specific enzymes contributing to this process. This is well illustrated by the effector BPE123 which has been shown to control host metabolism by interacting and recruiting α-enolase to the vacuolar membrane, enhancing its activity, a process essential for intracellular replication[46]. However, our data do not currently support the induction of ribophagy or any form of degradative autophagy. We have found that Bif lacks features of degradative autophagosomes. Therefore, *Brucella* may be highjacking a Beclin-dependent stress response to control the localisation of nuclear proteins, ribosomal biogenesis or host protein synthesis. Interestingly, additional nucleolar targets connected to ribosomal biogenesis were identified as potential interacting partners of NyxA in the yeast two-hybrid screen. CEBPZ, also known as Noc1, is involved in both pre-rRNA processing and maturation of pre-ribosomes[47]. It has been shown to participate in the intra-nuclear transport of the 60S subunits[47].

ARL6IP4 is an RNA splicing regulator shown to interact with core components of the 60S ribosomal subunit[48]. Therefore, it is possible that the Nyx effectors target large complexes of proteins associated with the maturation of the 60S ribosomal subunit, retaining them in the vicinity of multiplying bacteria. Alternatively, these may simply constitute markers of an autophagy-derived process elicited by extensive intracellular bacterial multiplication and maintained by the nucleoplasmic retention of SENP3 by NyxA and NyxB.

## Methods

### Cells, culture conditions and drug treatments

HeLa (CCL-2), HEK293T (CRL-3216) and RAW 264.7 (TIB-71) cell lines were obtained from ATCC and were grown in DMEM supplemented with 10% of foetal calf serum. We have not authenticated these ATCC cell lines. Immortalised bone marrow-derived macrophages[49] cell line originally derived from C57BL/6J mice were provided by Thomas Henry (CIRI, Lyon, France) maintained in DMEM supplemented with 10% FCS and 10% spent medium from L929 cells that supply MC-CSF. All cells were routinely tested and were mycoplasma free. When indicated, cells were treated with 0.5 mM arsenite (Sigma) for 30 min to induce stress granules and P-bodies and treated with 5 μM thapsigargin (Sigma) 4 h to induce U-bodies.

### Brucella strains, cultures and infections

*B. abortus* 2308 was used in this study. Wild-type and derived strains were routinely cultured in liquid tryptic soy broth and agar. 50 μg/ml kanamycin was added for cultures of DSRed and 50 μg/ml ampicillin for strains expressing pBBR1MCS-4[50] (4HA and 3Flag constructs) or complemented strains (pUCTminiTn7T_Km)[51]. The list of all *Brucella* strains constructed is included in Supplementary Table 5.

For infections, eukaryotic cells were plated on glass coverslips 18 h before infection and seeded at $2 \times 10^4$ cells/well and $1 \times 10^5$ cells/well for 24 and 6 well plates, respectively. Bacterial cultures were incubated for 16 h from isolated colonies in TSB shaking overnight at 37 °C. Culture optical density was controlled at 600 nm. Bacterial cultures were diluted to obtain a multiplicity of infection (MOI) of 1:500 for epithelial cells and 1:300 for iBMDM in the appropriate cell culture medium. Inoculated cells were centrifuged at $400 \times g$ for 10 min to initiate bacterial-cell contact followed by incubation for 1 h at 37 °C and 5% $CO_2$ for HeLa cells and 15 min for iBMDMs. Cells were then washed three times with DMEM and incubated for a further hour with complete media supplemented with gentamycin (50 μg/mL) to kill extracellular bacteria. The gentamycin concentration was then reduced to 10 μg/mL by replacing the media. At the different time points, coverslips were fixed for immunostaining. For enumeration of bacterial colony forming units (CFU), cells were lysed in 0.1% Triton for 5 min and a serial dilution plated in tryptic soy agar.

### Brucella expressing vectors

**TEM vectors.** DNA fragments coding for NyxA and NyxB were obtained by PCR amplification from the *B. abortus* 2308 genome, digested with XbaI and cloned into pFlagTEM1[52]. After verification by sequencing, plasmids were introduced into *B. abortus* 2308 or Δ*virB9* by electroporation.

**Deletion mutants.** *B. abortus* 2308 knockout mutant Δ*nyxA* and Δ*nyxB* were generated by allelic replacement. Briefly, upstream and downstream regions of about 750 bp flanking each gene were amplified by PCR (Q5 NEB) from the *B. abortus* 2308 genomic DNA. An overlapping PCR was used to associate the two PCR products and the Δ*nyxA* or Δ*nyxB* fragments were digested and cloned in an EcoRI suicide vector (pNPTS138)[53]. These vectors were then electroporated in *B. abortus* and transformants were selected using the kanamycin resistance cassette of the pNPTS138 vector. The loss of the plasmid concomitant with either deletion or a return to the wild-type phenotype

was then selected on sucrose, using the *sacB* counter-selection marker also present on the vector. Deletant (Δ) strain was identified by PCR.

**Complementing strains.** The complementing strain was constructed by amplifying either *nyxA* or *nyxB* and their corresponding promoter region (500 bp upstream) with the PrimeStar DNA polymerase (Takara). Insert and pmini-Tn7 were digested with SacI and BamHI and ligated overnight. Transformants were selected on kanamycin 50 μg/mL and verified by PCR and sequencing. To obtain the complementing strain the Δ*nyxA* or Δ*nyxB* mutants were electroporated with pmini-Tn7-*nyxA* or -*nyxB*, respectively, with the helper plasmid pTNS2. Electroporants were selected with kanamycin 50 μg/mL and verified by PCR.

**4HA and FLAG vectors.** To construct the 4HA-tagged vectors, the *nyxA* and *nyxB* genes were sub-cloned in plasmid pMMB 207c[20]. The *nyxA* and *nyxB* genes were previously amplified from pGEM-T-easy-*nyxA* or -*nyxB* plasmid. The primers used to amplify *nyxA/B* have KpnI and HindIII restriction sites on the forward and reverse primers, respectively. The PCR products were cleaved by these restriction enzymes to be inserted into the same restriction sites of pMMB207c downstream of the 4HA tag.

Subsequently, *4HA-nyxA* or -*nyxB* were amplified with primers having the SacI and SpeI restriction sites on the forward and reverse primers, respectively. The PCR products were cleaved by these restriction enzymes to be inserted into the same restriction sites of pBBR1MCS-4.

**MAG mutants.** The mutants NyxA^MAG and NyxB^MAG were obtained from pET151-*nyxA* or -*nyxB* and pmini-Tn7-*nyxA* or -*nyxB* using QuickChange Site-Directed Mutagenesis.

The mutants NyxA^MAG and NyxB^MAG correspond to three mutated amino acid residues.

These three mutations were introduced sequentially by mutating residues Y62R, D76R, E78R and Y66R, D80R, and E82R for NyxA and NyxB, respectively.

The resulting vectors were either introduced in *B. abortus*, or *B. abortus* expressing DSRed or GFP. All the primers used are listed in Supplementary Table 6 and all constructs were verified by sequencing.

### Eukaryotic expression vectors

The NyxA and NyxB constructs were obtained by cloning in the gateway pDONR™ (Life Technologies) and then cloned in the pENTRY Myc or HA. The NyxA^MAG and NyxB^MAG constructs were obtained by site-directed mutagenesis as described above.

The *4HA-nyxA* and *4HA-nyxB* genes were amplified from the plasmid pBBR1MCS4-4HA-*nyxA* or -*nyxB*. The primers used to amplify *4HA-nyxA* or -*nyxB* have the BamHI and EcoRI restriction sites in forward and reverse primers, respectively. The PCR products were cleaved by these restriction enzymes to be inserted into the same restriction sites of pcDNA3.1. All the primers used are listed in Supplementary Table 6.

### Bacterial expression vectors

NyxA and NyxB were amplified by PCR and inserted into the pET151D topo vector following the manufacturer's procedure (Invitrogen) to obtain His-NyxA and His-NyxB. An additional V5 tag is also present. His-NyxA^MAG and His-NyxB^MAG were obtained by site-directed mutagenesis as described above. For expression and purification of SENP3$_{7-159}$, a vector with *E. coli* codon-optimised SENP3 was obtained from Thermofisher and used as a template. SENP3$_{7-159}$ was cloned into pRSF-MBP vector. This vector corresponds to pRSFDuet-1 (Novagen) but was modified to insert 6xHis-MBP and a protease recognition site for tobacco etch virus (TEV) protease.

All the primers used are listed in Supplementary Table 6.

## Immunofluorescence microscopy

At the indicated time points, coverslips were washed twice with PBS, fixed with AntigenFix (MicromMicrotech France) or PFA 3.2% (Electron Microscopy Sciences) for 20 min and then washed again four times with PBS. To detect translocated HA and Flag-tagged effectors, permeabilisation was carried out with a solution of PBS containing 0.3% triton for 10 min to minimise permeabilisation of BCV membranes. This was followed by blocking with PBS containing 2% bovine serum albumin (BSA), and 10% horse serum. Primary antibodies diluted in 2% BSA and 10% HS were then incubated at 4 °C overnight in the blocking solution. Subsequently, the coverslips were washed twice in PBS and incubated for 1 h with secondary antibodies. Finally, coverslips were washed twice in PBS and once in ultrapure water. Lastly, they were mounted on a slide with ProLongGold (Life Technologies). The coverslips were visualised with a Confocal Zeiss inverted laser-scanning microscope LSM800 and analysed using ImageJ[54] software and final figures were assembled with FigureJ[55]. Note that neither blocking nor antibody solutions did not contain Triton (only the permeabilisation step). It is important to point out that occasional intravacuolar labelling was observed in weak DSRed bacteria, perhaps indicative of fragile BCVs. The same protocol was done for NUFIP1.

For all other antibodies, when used in the absence of 4HA or 3Flag for translocated effectors, cells were permeabilised with a solution of PBS containing 0.5% triton for 20 min followed by blocking for 20 min in a solution of PBS containing 2% bovine serum albumin (BSA), 10% horse serum, and 0.3% triton. Coverslips were then incubated either at 4 °C overnight with primary antibody diluted in the blocking solution containing 0.3% Triton. Successful labelling was also achieved with 3 h incubation at RT. The subsequent steps were identified as described above. The commercial NVL antibody (16970-1-AP) was not stable when stored at −20 °C for prolonged periods but gave similar results to the homemade NVL antibody (from M. Nagahama). Both homemade NVL/RPL5 antibodies and commercial anti-NVL and SENP3 antibodies were tested in siRNA-depleted cells for specificity.

For cytosolic labelling, SENP3 cells were washed three times with PBS for 5 min each, fixed with chilled 4% paraformaldehyde for 20 min and then washed three times with PBS. Cells were permeabilized with 0.01% digitonin (Sigma) for 2 min and then washed three times in PBS. Cells were blocked for 60 min with 10% goat serum, 2% BSA in PBS and washed three times in PBS. SENP3 antibody was then incubated overnight at 4 °C in PBS and secondary antibody staining was done at 37 °C for 1 h.

Mitotracker Red CMXRos® (Life Technologies) was used at 100 nM for 45 min and ER tracker Red® (Thermo) at 1 μM for 45 min before fixation with 3% paraformaldehyde.

For the proximity ligation assay, we used the DuoLink® In situ mouse/rabbit kit (Merk) and followed the manufacturer's instructions. For primary antibodies, we used the protocol described above to detect translocated FLAG-tagged effectors. When specified, the cytosolic SENP3 permeabilization protocol was used.

All fixations, dilutions and sources for each antibody are indicated in Supplementary Table 7.

## Transfections and siRNA

All cells were transiently transfected using Fugene® (Promega) overnight, according to the manufacturer's instructions. HeLa cells were seeded 18 h prior to experiments at $2 \times 10^4$ cells/well for 24-well plates. siRNA experiments were done with Lipofectamine® RNAiMAX Reagent (Invitrogen) according to the manufacturer's protocol. Cells were seeded at $2 \times 10^4$ cells/well. Importantly, siRNA depletion of SENP3 was done by treatment with 5 nM siRNA every day for 72 h. Some cell detachment was observed following the 3 days of siRNA treatment. Results were confirmed with an alternative siRNA from HorizonDiscovery, efficient after 2 days of silencing. For siBeclin depletion and PIAS3, 2 days of treatment was used with 10 nM. A scrambled

siControl was always included in all experiments. siRNA of NVL and RPL5 treatment were done for either 48 or 72 h and resulted in significant cytotoxicity and therefore were not used in the context of infection. All siRNA used are described in Supplementary Table 6.

## *Legionella* infection

*Legionella pneumophila* strain Paris was cultured in *N*-(2-acetamido)−2-aminoethanesulfonic acid (ACES)-buffered yeast extract broth or on ACES-buffered charcoal-yeast (BCYE) extract agar[56]. The human monocyte cell line (THP-1) was cultured and infected as previously described[57].

For immunofluorescence analyses, cells were fixed in 4% paraformaldehyde, permeabilised with PBS-triton 0.5% and stained with 4-6-diamidino-2-phenylindole (DAPI), Phalloidin and primary anti-NVL antibody (16970-1-AP). Immunosignals were analysed with a Leica SP8 microscope at ×63. Images were processed using ImageJ software.

## Pulldown assays

For pulldown experiments with two purified proteins, 50 μg of His tag recombinant protein was incubated with recombinant protein for 2 h at 4 °C, then incubated within a gravity flow column (Agilent) containing 80 μl Ni-NTA agarose beads (Macherey-Nagel) during 1 h beforehand washed in water and pre-equilibrated in equilibrium buffer 20 mM Tris−HCl pH 7.5, 250 mM NaCl. The column was washed successively three times in equilibrium buffer supplemented with 25 mM imidazole, three times in equilibrium buffer and eluted in equilibrium buffer supplemented with 500 mM imidazole. Proteins eluted were separated by SDS−PAGE, transferred to a PVDF membrane, incubated with specific primary antibodies for 1 h and detected with horseradish peroxidase (HRP)-conjugated secondary antibodies by using ClarityTM Western ECL Blotting Substrate (Bio-Rad).

For pulldown assays between cell extract and purified Nyx and MAG mutants, HeLa cells were seeded at $5 \times 10^5$ in a 10-cm cell culture dish and incubated overnight in a 37 °C humidified atmosphere of 5% $CO_2$. 22 h after incubation, cells were washed in ice-cold PBS, harvested and resuspended in 200 μl of RIPA buffer (Sigma) supplemented with phenylmethylsulfonyl fluoride (Sigma) and protease inhibitor cocktail (Roche). Extracts were incubated on ice for 15 min with periodical pipetting, then centrifuged at $16,000 \times g$ at 4 °C for 20 min to pellet cell debris. The supernatant was transferred to a fresh tube. 50 μg of His tag recombinant protein was incubated with 80 μl Ni-NTA agarose beads (Macherey-Nagel) within a gravity flow column (Agilent) for 1 h beforehand washed in water and pre-equilibrated in equilibrium buffer 20 mM Tris− HCl pH7.5, 250 mM NaCl. The cell extract was incubated within the column (containing His tag recombinant protein and Ni-NTA agarose beads) for 1 h at 4 °C under agitation. The column was washed successively three times in equilibrium buffer supplemented with 25 mM imidazole, three times in equilibrium buffer and eluted in equilibrium buffer supplemented with 500 mM imidazole.

## Co-immunoprecipitation

HeLa cells were cultured in 100 mm × 20 mm cell culture dishes at $1.5 \times 10^6$ cells/dish overnight. Cells were transiently transfected with the JetOPTIMUS (Polypus) with a total of 10 μg of DNA/plate of either pGFP or GFP-SENP[58] simultaneously with either myc-NyxA or myc-NyxB for 18 h (media was replaced 4 h after transfection). On the ice, after two washes with cold PBS, cells were collected with a cell scraper and centrifuged at $80 \times g$ at 4 °C for 5 min. Cell lysis and processing for co-immunoprecipitation were done as described with the GFP-Trap (Chromotek).

## Quantification of SENP3 localisation

Colocalisation analysis for the transfected HeLa cells was performed with a custom ImageJ/Fiji[59]-based macro (http://c2bp.ibcp.fr/CL_ArthurL_coloc_2-corr.ijm), that segmented the nuclei and the nucleoli

of the cells in each image, classified the cells in two classes according to the intensity of HA-NyxA/NyxB, and then measured in the areas of each nucleus the Pearson correlation coefficients (by calling the plugin Coloc2–https://github.com/fiji/Colocalisation_Analysis) between the signal of SENP3 and the nucleolin as well as between the signals of SENP3 and HA-NyxA/NyxB. For each nucleus, the ratio between the mean intensity of the SENP3 signal in the nucleoli and the mean intensity of the SENP3 signal outside the nucleoli is also calculated. The details can be found in the source code and the comments of the macro.

For the *Brucella-infected* HeLa cells, a pipeline in the software CellProfiler[60] was used to measure the Pearson correlation coefficient between the signal of SENP3 and the nucleolin, in the nuclei of the cells. The cells were also classified into two classes according to the intensity of the *Brucella* DSRed signal in the perinuclear area. The pipeline can be downloaded here http://c2bp.ibcp.fr/arthur2.cpproj.

### TEM1 translocation assay
RAW cells were seeded in 96-well plates at $1 \times 10^4$ cells/well overnight. Cells were then infected with an MOI of 500 by centrifugation at 4 °C, 400×*g* for 5 min and 1 at 37 °C 5% $CO_2$. Cells were washed with HBSS containing 2.5 mM probenicid. Then CCF2 mix (as described in the Life Technologies protocol) and probenicid were added to each well and incubated for 1.5 h at room temperature in the dark. Cells were finally washed with PBS, fixed using 3.2% PFA and analysed immediately by confocal microscopy (Zeiss LSM800).

### Protein expression and purification
*E. coli* BL21-DE3-pLysS bacteria were transformed with the expression vectors, grown in lysogenic broth (LB) media (Sigma-Aldrich) to $OD_{280} = 0.6$ and expression was induced with 1 mM IPTG at 37 °C for 3 h. Cells were After harvested by 15 min of centrifugation at 5000×*g* and resuspended in lysis buffer 20 mM Tris pH 8, 150 mM NaCl, 5% glycerol, 0,1% Triton. Disruption cell was achieved with sonication after the addition of an antiprotease EDTA-Free cocktail (Roche) and 30 U/ml benzonase (Sigma-Aldrich). Cell debris was removed by centrifugation for 30 min at 20,000×*g* at 4 °C. Recombinant protein was purified by chromatography using a Nickel-loaded Hitrap Chelating HP column (GE Healthcare). Unbound material was extensively washed using Tris 20 mM pH 8, NaCl 300 mM, 25 mM Imidazole, 5 mM β-mercaptoethanol, and 10% glycerol. An additional washing step with 2 column volumes of 1 M NaCl was done before elution of NyxB over a 25–500 mM gradient of imidazole over 8 column volumes. Peak fractions were pooled and the His tag was cleaved with TEV protease (500 μg/20 mg of eluted protein) in presence of 1 mM DTT and 0.5 mM EDTA in overnight dialysis buffer 20 mM Tris pH 8, 150 mM NaCl. NyxB was further purified by size-exclusion chromatography (Superdex 200 HiLoad 16/600, GE Healthcare) equilibrated in 20 mM Tris pH 8, 150 mM NaCl, and 5% glycerol. The purity of the sample was assessed by SDS–PAGE. Freshly purified NyxB was concentrated at 21 mg/ml on 3 kDa Amicon Ultra concentrators (Millipore). SeMet-NyxB was produced in M9 minimum medium and purified as above, to a final concentration of pure SeMet-NyxB of 24 mg/ml.

### Crystallisation and data collection
The screening was conducted using a Mosquito workstation (TTP Labtech) on commercial crystallisation solutions with the sitting-drop vapour diffusion technique, against a protein solution. All crystallisation trials were performed at 19 °C and visualised on RockImager 182 (Formulatrix). Crystals of native NyxB were obtained with 21 mg/ml NyxB in 25% PEG4000, 0.2 M $CaCl_2$, 100 mM Tris pH 8. Crystals were frozen in a reservoir solution supplemented with 15% Gly. Diffraction data were collected at the European Synchrotron Radiation Facility (ESRF) Beamline line ID23EH1 and crystals diffracted up to 2.5 Å resolution in space group $P6_122$, with 12 molecules per asymmetric unit.

Crystals of Se-Met NyxB were obtained using a reservoir solution containing 2.6 M NaFormate and crystals were cryoprotected in 2.8 M Naformate supplemented with 10% glycerol. Data were collected at SOLEIL on beamline Proxima-2 from a single crystal that diffracted up to 3.7 Å resolution and belonged to the space group $P6_222$, with 2 molecules per asymmetric unit. Diffraction data were processed using XDS[61] and with AIMLESS[62] from the CCP4 programme suite[63]. Data collection statistics are indicated in Supplementary Table 2.

The structure was solved using the single anomalous dispersion method on Se-Met crystals using AutoSol from the Phenix programme suite[64]. An excellent experimental electron density map enabled us to manually build an initial model. The resulting model was then used for molecular replacement with data from native NyxB crystal using Phaser. Twelve monomers were positioned and the resulting electron density map was then subjected to the AutoBuild programme, part of the Phenix programme suite. Model building was completed with sessions of the manual model building using Coot combined with model refinement using Phenix[65]. The final model was refined to a final $R_{work}/R_{free}$ of 0.21/0.24 with excellent geometry (Supplementary Table 2). The coordinates and structure factors of NyxB have been deposited in the protein DataBank with code 8AF9. Figures were generated with Pymol, Molecular Graphics System, Version 2.0 Schrödinger, LLC.

### Size exclusion multi-angle light scattering
Size-exclusion chromatography (SEC) experiments coupled to multi-angle laser light scattering (MALS) and refractometry (RI) were performed on a Superdex S200 10/300 GL increase (GE Healthcare) for NyxA and a Superdex S75 10/300 GL (GE Healthcare) for NyxB. Experiments were performed in buffer 20 mM Tris pH 8, 150 mM NaCl, and 5% glycerol. 100 μl of proteins were injected at a concentration of 10 mg ml$^{-1}$. On-line MALS detection was performed with a miniDAWN-TREOS detector (Wyatt Technology Corp., Santa Barbara, CA) using a laser emitting at 690 nm and by refractive index measurement using an Optilab T-rex system (Wyatt Technology Corp., Santa Barbara, CA). Weight averaged molar masses (Mw) were calculated using the ASTRA software (Wyatt Technology Corp., Santa Barbara, CA).

### Small-angle X-ray scattering
SAXS data were collected for NyxA and NyxB on BioSAXS beamline BM29, ESRF using an online size-exclusion chromatography setup. 50 μl of protein (10 mg ml$^{-1}$) were injected into a size-exclusion column (Agilent BioSec-3) equilibrated in 50 mM Tris, pH 8.0, 200 mM NaCl. Images were acquired every second for the duration of the size-exclusion run. Buffer subtraction was performed by averaging 20 frames on either side of the peak. Data reduction and analysis were performed using the BsxCuBE data collection software and the ATSAS package[66]. The programme AutoGNOM was used to generate the pair distribution function ($P(r)$) and to determine $D_{max}$ and $R_g$ from the scattering curves ($I(q)$ versus $q$) in an automatic, unbiased manner. Theoretical curves from the models were generated by FoXS[67]. Ab initio modelling was performed with GASBOR[68].

### Yeast two-hybrid
NyxA was cloned into pDBa vector, using the Gateway technology, transformed into MaV203 and used as bait to screen a human embryonic brain cDNA library (Invitrogen). Media, transactivation test, screening assay and gap repair test were performed as described[69–71].

### Graphs, statistics and reproducibility
For microscopy counts of individual cells, super plots were used[72]. All data sets were tested for normality using the Shapiro–Wilkinson test. When a normal distribution was confirmed we used a One-Way ANOVA test with a Tukey's correction for multiple comparisons. For two independent variables, a Two-Way ANOVA test was used. For data sets

that did not show normality, a Kruskal–Wallis test was applied, with Dunn's correction. All analyses were done using Prism Graph Pad 7.

For all microscopy, data were obtained from at least three independent experiments, and for all western blots data was reproduced in two independent experiments.

## Reporting summary

Further information on research design is available in the Nature Portfolio Reporting Summary linked to this article.

## Data availability

All data generated or analysed during this study are included in this published article (and its supplementary information files) or are provided as a Source Data file. The structure is available as PDB 8AF9 Source data are provided with this paper.

## Code availability

The pipeline code for image analysis of SENP3 in infected cells can be downloaded as a Cell Profiler project at http://c2bp.ibcp.fr/arthur2.cpproj. This file can be opened with the software CellProfiler 3.0.0 (available at https://cellprofiler.org/, following the instructions at https://cellprofiler.org/getting-started) and contains all the steps used to analyse the images.

Colocalisation analysis for the transfected HeLa cells was performed with a custom ImageJ/Fiji macro (http://c2bp.ibcp.fr/CL_ArthurL_coloc_2-corr.ijm) as described in the "Methods" section. All the information can be found in the macro.

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

## Acknowledgements

We would like to thank Rémi Lagorce for help with the purification of the N-terminus of SENP3, Inès Bordin for the optimisation of the expression conditions of NyxB, Marie-Pierre Candusso and Mégane Guinot from the Protein Science Facility of SFR Biosciences for help with protein purification of NyxB and crystallography experiments. We are grateful to Gunnar Schroeder (Queen's University of Belfast) for the pMMB 207c, Renée Tsolis (University of California, Davis) for the pFlagTem, Matthias Faure (CIRI, France) for the LC3-GFP expression vector, Xavier de Bolle (University of Namur) for the pNPTS138 and Jean Celli (Washington State University) for the pUCTminiTn7T_Km. We are also thankful for the HEK NUFIP KO cells gently provided by Heeseon An and Wade Harper (Harvard University, USA) and for the iBMDM by Thomas Henry (CIRI, France). The LAMP1

and Myc antibodies developed by Bishop, J.M. were obtained from the Developmental Studies Hybridoma Bank, created by the NICHD of the NIH and maintained at the University of Iowa. GFP-SENP3 was a gift from Mary Dasso (Addgene plasmid #34554; RRI-D:Addgene_34554). CB lab was funded by the Institut Pasteur and ANR-10-LABX-62-IBEID for the *Legionella* experiments. We acknowledge the contribution of the SFR Biosciences (UAR3444/CNRS, US8/Inserm, ENS de Lyon, UCBL) facilities: Protein Science Facility for mass spectrometry and crystallography experiments and the Plateau Technique Imagerie/Microcopie (PLATIM) for all microscopy studies. We thank staff scientists from beamlines Proxima-2 (synchrotron SOLEIL), ID23EH1 (ESRF), and BM29 (ESRF) for assistance with data collection. These effectors were discovered under the ERA-Net Pathogenomics grant and the remaining work funded by ANR-15-CE15-0011-01 attributed to Suzana Salcedo. The work was completed with the ANR SNAPshot ANR-21-CE15-0024 attributed to Suzana Salcedo. We also thank Steve Garvis and Stéphane Méresse for critically reading the manuscript. Jean-Paul Borg is a scholar of the Institut Universitaire de France.

## Author contributions

A.L. carried most of the initial experimental work with *Brucella* (cell biology, infection studies and biochemistry). A.B. did the assessment of *Brucella* replication with siSENP3 and discovered NVL/RPL5 delocalisation and Supplementary Fig. 13. She carried out the majority of the experiments required for the revision of the manuscript and therefore both authors contributed equally and agreed to share authorship. She also carried out the optimisation of the protein purification of NyxA for the MALS. T.L.S.L. discovered these effectors and showed their translocation and nuclear targeting in transfection. LCV and FG did some of the biochemistry requested during the revision. CL coded the plug-ins for the quantification of SENP3 and assisted with image analysis. C.Be. performed and analysed SAXS experiments, M.R. and C.Bu. the *Legionella* experiment, F.L. and J.P.B. did the Y2H. This work was started in the lab of J.P.G. who contributed to the discovery of this effector. M.N. provided tools and input regarding NVL, RPL5 and ribophagy. V.G.C. and L.T. supervised and performed the MALS, the crystallography screens and the diffraction data collection. L.T. solved, analysed the structure of NyxB, designed the MAG mutant and wrote the structural part of the manuscript. S.P.S. directed, funded and supervised the vast majority of the work, and wrote the manuscript with A.L. and A.B. All other authors read and approved the manuscript.

## Competing interests

The authors declare no competing interests.
