## [Peer Review File · Nature Communications]

Brucella effectors NyxA and NyxB target SENP3 to modulate the subcellular localisation of nucleolar proteinsReviewer #1 (Remarks to the Author):

This study reports the discovery and characterisation of two closely related effector proteins of the intracellular bacterial pathogen *Brucella abortus*, which represent a new family of bacterial nucleomodulins that alter the cellular localisation of nucleolar proteins. Using a robust combination of genetic, cellular and structural approaches, the authors convincingly show that NyxA and NyxB are delivered into host cells during infection, target the nucleolar SUMO protease SENP3 and induce cytoplasmic structures (*Brucella*-induced foci or Bifs) that contain the ribosomal biogenesis-associated nucleolar components and regulators NVL and RPL5 and the ribophagy receptor NUFIP. They further show that Bif formation requires the autophagy protein Beclin 1 and is down regulated by SENP3, suggesting a ribophagic process triggered by mislocalisation of SENP3 by NyxA/B.

This manuscript includes an extensive set of experiments that are well designed and executed and support the specific conclusions drawn by the authors. Understanding how bacterial pathogens modulate nuclear functions is a current topic of research in infectious diseases and very little is known about the functions of *Brucella* Type IV secretion effectors. In this context, the findings presented here provide important advances to several areas of research in bacterial pathogenesis. I remain, however, confused by how the authors think NyxA and NyxB operate, so I would recommend clarifying this aspect of the manuscript.

Major points:

1. Although most of the data is strong and convincing, I am confused about the proposed overall mode of action of Nyx effectors on SENP3 with regard to Bifs and enhanced bacterial growth. While NyxA (and NyxB to a certain extent) causes mislocalisation of SENP3 and both clearly induce Bif formation in a manner dependent on binding to SENP3, do the authors claim that Nyx effectors retain SENP3 in the nucleus to prevent it from regulating ribophagy? If so, why are Nyx effectors detected in Bifs? Is SENP3 also recruited to Bifs, which are NyxA/B-induced structures? In infected SENP3 KD cells, which form an increased number of Bifs, do NyxA and NyxB localize to the enhanced cytosolic structures? I recommend that the authors clarify these points.

2. The authors present evidence supporting a ribophagic nature of Bifs, in that they are positive for the ribophagy receptor NUFIP and their formation requires Beclin1. Yet, I think this important finding needs to be further substantiated with evidence for additional autophagic markers in Bifs: based on the data presented, is Beclin 1 detected on Bifs? Given NUFIP presence on Bifs, is LC3B (Which NUFIP directly binds) detected? Are lysosomal markers, such as LAMP1, present on Bifs, as would be expected for degradative ribophagy? The lack of detection of ubiquitin using the FK2 antibody is interesting but not really pursued. It may reflect a ubiquitin-independent process or ribophagic structures that are matured enough for ubiquitin to be degraded.

3. If Bifs represent ribophagy events providing nutrients to enhance bacterial growth, as speculated in the discussion, shouldn't SENP3 depletion also promote replication and not impair it, since it negatively regulates Beclin1-dependent ribophagy? Or are there additional effects of SENP3 depletion to the cells that independently affect bacterial multiplication and confound the interpretation? Also in this context, does Beclin1 KD affects optimal replication at 48 h pi?

Minor points:

1. Line 3: Based on the slight decrease in multiplication presented in Fig. 2C, it is a bit excessive to claim that SENP3 is "essential" for intracellular replication. I suggest replacing it with "contributes to optimal growth" or an equivalent statement. Similarly, I would tone down the sentence "...key role of SENP3 in the infectious process..." (lines 441-442).

2. Lines 85 and 455: The Brucella T4SS is characterised by the absence of the VirD4 component, so perhaps naming it "VirB T4SS" instead of "VirB/D" is more appropriate.

3. Line 92: ...with SENP3...

4. Fig. 1B-C. please add scale bars to the image and figure legend.

5. Fig 1E and S3A, lines 177-181: the pulldown experiments show a very small amount of the untagged proteins coming down in the eluates, so the interactions between purified Nyx proteins is not very convincing, despite the medium affinity Kd measured separately. There seems to be some methodological limitations to the interaction when using purified proteins. Could another technique be used, such as co-expression in cells followed by co-IP? Ultimately, showing an interaction between the two effectors does not seem to add much to the manuscript, unless the authors suggest that heterodimerization is functionally meaningful.

6. Lines 207-214: As above, given the limitations with obtaining purified full length SENP3, could a co-IP between Nyx and full length SENP3 in mammalian cells also confirm the interaction between these proteins? While not showing a direct interaction, it would convincingly complement the Y2H data.

7. Fig. S8 and lines 349-353: Not to sound repetitive or obsessive, but the pull-down approach may not be the best methodology to use here, as an unusual cytosolic complex containing SENP3 and NVL may only form upon infection or ectopic expression of the Nyx effectors. A co-IPs of HA-Nyx effectors would be more appropriate in this instance and may reveal that SENP3 and NVL are part of a same Nyx-induced complex.

8. Line 368: ...We hypothesised if.... Please correct the grammar of this sentence.

9. Line 379: ...co-label NVL and RPL5....

10. I would suggest adding the quantification shown in Fig S13A to Fig 5F to emphasize this important piece of data.

11. Lines 408-412: Please provide a Western blot of Beclin1 KD levels.

12. Line 415-417: the reasoning in this sentence is a bit awkward: one would expect a focus on SENP3 activity in Bifs, and not "elsewhere in the cell". Please clarify.

13. Lines 506-508: This sentence implies that NyxA/B act in the nucleolus to retain SENP3 from regulating ribophagy in the cytoplasm, but they are detected in Bifs. This is linked to major point #1.

14. Line 503: "overflow valve" could use a more scientific description.

15. Line 515: "Brucella could be eating the cell's ribosomes" could be replaced with a more scientific description.

16. Lines 148 and 518: I would recommend using a more descriptive qualifier than "nicely".

Jean Celli

Reviewer #2 (Remarks to the Author):

In this manuscript, Louche et al. have demonstrated that *Brucella* effectors NyxA and NyxB interact with SENP3 in the nuclei, and these interactions seem to directly cause endogenous nucleolar SENP3 translocate to the nucleoplasm. However, deletion of both genes in *Brucella* does not seem to be sufficient to completely recover SENP3 nucleolar localization (suggesting the involvement of other unknown *Brucella* effector(s)). Moreover, SENP3 translocation of this kind also seemed to occur upon *B. abortus* infection, which also resulted in the cytoplasmic accumulation of the nucleolar protein NVL in *Brucella*-induced foci (Bif) in a manner dependent on Nyx effectors. SENP3 depletion seemed to affect *B. abortus* replication in HeLa cells, increase Bif, and induced the cytoplasmic accumulation of the nucleolar protein NVL in infected HeLa cells but not in non-infected controls. Interestingly the ribosomal protein RPL5, the Nyx effectors and the nucleo-cytoplasmic shuttling protein NUFIP1 known as a receptor for ribophagy were detected in the *Brucella*-induced NVL foci. (It is claimed that) Beclin1 knockdown seems to reduce Bif. Furthermore, *L. pneumophila* infection does not appear to result in cytoplasmic NVL accumulation so the phenomenon specifically occurs upon *B. abortus* infection. Although these observational findings are interesting and potentially important, the authors' investigations are largely based on their speculations. Therefore, this study appears to be at an early stage, and substantial efforts are needed to figure out: i) if the mediating role(s) for SENP3 in these cellular events are dependent on its deSUMOylation activity; ii) if those proteins involved are deSUMOylating substrates for SENP3; iii) if SENP3-mediated autophagies including ribophagy or SENP3 mediated deSUMOylation of certain proteins results in Bif upon *B. abortus* infection.

My comments are listed as follows:

Major points

1) In the manuscript It has been speculated that *Brucella* effectors-SENP3 interaction negatively regulates deSUMOylation of Beclin-1 by SENP3 to cause NUFIP1-mediated ribophagy, in turn result in cytoplasmic accumulation of the nucleolar protein NVL in Bif that containing RPL5 and NUFIP1 upon *B. abortus* infection. To gain mechanistic insights as to how those cellular events really happen and how they are really associated, the following questions need to be answered:

Does expression of the two effectors or *B. abortus* infection increase in Beclin1 SUMOylation levels in the cells used?

Does SENP3 deSUMOylate Beclin-1 in non-infected cells?

Does *B. abortus* infection causes general autophagy and/or selective autophagy types, including ribophagy (by conducting LC3 conversion and ribophagic reporter assays (1)), through regulating SUMOylation status of Beclin1?

Does expressing the effectors or *B. abortus* infection cause changes in Beclin1 expression levels in the cells used?

Is NUFIP1 involved in *B. abortus*-induced Beclin1-dependent ribophagy (if any)?

Is Bif resulted from *B. abortus*-induced Beclin1-dependent ribophagy (if any)?

2) As a potent autophagy inducer, Rapamycin (originally isolated from bacterium *Streptomyces hygroscopicus*) is known to induce nucleolar SENP3 translocate to the nucleoplasm via inhibiting mTOR pathway. It would be necessary to know if expression of the two effectors or *B. abortus* infection affects mTOR signaling.

3) NVL, RPL5 and NUFIP1 are identified as SUMO-2 target proteins identified by mass spectrometry (2): K208, K164/K188/K221, and K229 are SUMO conjugation sites in NVL, RPL5, and NUFIP1, respectively. Therefore sequestering or depletion of SENP3 would enhance their SUMOylation and that would result in their translocation(s).

In this respect, the following questions need to be answered:

Does expression of the two effectors or *B. abortus* infection increase in SUMOylation levels of these 3 proteins in the cells used?

Does SENP3 depletion increase in SUMOylation levels of these 3 proteins in the cells used?

**Does SUMOylation regulate the translocation of NVL, RPL5 or NUFIP1?
Does sequestering/depletion of SENP3 initiate the occurrence of multiple SUMO conjugations simultaneously at spatially related groups of target proteins such as NVL, RPL5, NUFIP1 and possibly Beclin-1 to induce autophagy/ ribophagy upon B. abortus infection?**

4) RPL5 gene seems to be regulated by SUMOylation (3). Therefore, it is necessary to examine if B. abortus infection induces changes in the levels of free SUMOs and SUMO conjugation in the cells used in this study.

5) Lack of mechanistic explanation regarding the discrepancy in subcellular locations of the Nyx receptors differentially tagged (i.e., HA/Myc vs 4HA/3Flag).

6) Lack of mechanistic explanation why SENP3 depletion increased the appearance of cells with "1-5" and "6-10" bacteria 48 h post infection (Figure 2C).

7) Lack of mechanistic explanation regarding why "NyxA did not form crystals in any of the conditions tested" for solving its structure.

8) Lack of mechanistic explanation regarding the observation that the authors stated on the page 13: "we could also observe a statistically significant increase of SENP3 in the nucleoli in cells infected with the Δ nyxB strain compared to cells infected with wild-type B. abortus, although to a lesser extent than what we observed for Δ nyxA."

9) Non-infected controls are needed for Figure 5A and Supplementary Figure 9;

10) On the page 7, experimental evidence is needed for the authors' statement: "This hypothesis was not confirmed in infected cells as effectors tagged with a single HA epitope were not detectable at any of the time-points analysed."

11) On the page 9, amino acid sequence details are needed for the authors' statement: "This region encodes two basic amino acid stretches predicted as nucleolar localisation sequences..."

12) On the page 11, the authors stated, "we could only detect a small amount of endogenous SENP3 in the cell extract with our antibody". According to the Supplementary Information, the antibody refers to the one (D20A10) from Cell Signaling Technology. Is possible to use an alternative antibody to further verify the finding?

13) Lack of sequence details of non-targeting siRNA as well as siRNA targeting SENP3 or Beclin1: they do not appear to be included in the Supplementary Table 6.

14) No experimental evidence showing Beclin1 siRNA-mediated depletion (either western blotting or qPCR confirmation is needed.)

15) On the page 16, the authors stated, "Due to antibody incompatibility we could not co-label with NVL and RPL5 simultaneously". According to the supplementary table 7, they used two antibodies (rabbit and mouse origins, respectively) for NVL detection and one rabbit antibody for RPL5. So the incompatibility needs to be clarified and an alternative experiment needs to be conducted.

16) On the page 18, the authors stated, "infection with L. pneumophila did not result in NVL cytoplasmic accumulation". Is it possible to introduce the Nyx effectors into the L. pneumophila to see if the bacteria can induce the appearance of NVL in the cytoplasm of infected cells in the presence of the Nyx effectors.

17) On the page 20, the authors demonstrate their siRNA-mediate SENP3 depletion (Figure 2B). However they stated, "Full depletion of SENP3...". Actually in this study, no knockout technology was used to fully deplete SENP3. So the statement is scientifically

inaccurate. Crucially to exclude potential siRNA off-target effect(s), it is necessary to use at least another siRNA to knock SENP3 down for verifying those effects observed in this study. Furthermore, to investigate if SENP3 action upon *B. abortus* infection is dependent on its deSUMOylation activity, SENP3 knockdown plus rescue experiments (with SENP3 wild type protein or inactive SENP3 mutant) should be essential in the cells used in this study.

Minor points

1) References are needed for:

The statement on the page 13, "One of the principal roles of SENP3 in the nucleoli is to regulate ribosomal biogenesis, specifically of the 60S ribosomal subunit."

The statement on the page 15, "we infected immortalised bone marrow-derived macrophages (iBMDMs), a well-established model of *Brucella* infection".

2) Few grammatical errors were spotted:

On the page 15, "if" (line 368) should be "that"

On the page 18, "that" (line 440) should be "which"

On the page 20, the sentence, "siRNA depletion of SENP3 occurs at an earlier stage of the replication than the effect induced upon NyxA/B translocation" (lines 484 and 485), is not easy for understanding. Please revise it.

On the page 22, "Alternative" (line 543) should be "Alternatively"

Reference:

1. An H, Harper JW. Systematic analysis of ribophagy in human cells reveals bystander flux during selective autophagy. *Nat Cell Biol* 20, 135-143 (2018).

2. Hendriks IA, D'Souza RC, Yang B, Verlaan-de Vries M, Mann M, Vertegaal AC. Uncovering global SUMOylation signaling networks in a site-specific manner. *Nat Struct Mol Biol* 21, 927-936 (2014).

3. Liu HW, et al. Chromatin modification by SUMO-1 stimulates the promoters of translation machinery genes. *Nucleic acids research* 40, 10172-10186 (2012).

Reviewer #3 (Remarks to the Author):

The MS of A. Louche et al. reports the identification of NyxA and NyxB, two new *Brucella* effectors that are translocated to the host cell where they target the SENP3 protease, promoting its delocalization from the nucleoli. These new effectors promote the formation of cytoplasmic structures enriched in the nucleolar proteins NVL, NUFIP1 and RPL5, in a process that is dependent on Beclin1 and negatively regulated by SENP3. The authors propose that NyxA/B, acting as two nucleomodulins, interact with SENP3 to enhance a ribophagy-like process that ultimately promotes *Brucella* intracellular proliferation.

This is an original, extensive, and relevant piece of work describing what seems a new family of bacterial effectors with nucleomodulatory functions. The authors succeed in identifying both effectors and their cellular target, which is per se a great task. However, they went beyond and resolved the crystal structure of NyxB what allow them to describe the acidic groove necessary for the interaction with SENP3, thus providing the biophysical basis of the interaction and confirming the Y2H and immunoprecipitation experiments. Then, they studied the effects of NyxA/B-SENP3 interaction discovering that it also promotes de delocalization of NVL and RPL5 from the nucleoli, accumulating

in cytoplasmic structures called Brucella induced foci (Bif) where the effectors are also located. The ribophagy receptor NUFIP1 is also recruited to the Bif in a NyxA/B dependent manner. So, the authors propose that NyxA/B target SENP3 to enhance cellular ribophagy as a way to promote the Brucella intracellular proliferation. The MS is well written and clear for the reader.. The main body contains the most important figures of the work, and the supplementary material is abundant and provides the necessary information for the full understanding of the article. The experiments are well designed and conducted in an elegant manner. In summary, this is an excellent and stimulating paper providing compelling evidence about a new family of bacterial effector with nucleomodulatory functions.

Minor comments:

Ln114. Ref 19 reports a Legionella effector not a "subgroup of Brucella candidate effectors". Please revised it.

Ln162-183. Does the authors have any hypotheses as to why NyxA and NyxB interact? Is this interaction important for targeting SENP3? Can they form a heterodimer? I would like to know what ideas the authors have about this.

Reviewer #4 (Remarks to the Author):

This manuscript presents a great job and is very worthy of publication. The author identified two new Brucella effectors, NyxA and NyxB, confirmed the target of these two effectors as SENP3 and discussed the functions during infection. The finding is novel and very helpful for understanding the mechanism of pathogen infections. The data as well as the explanations are solid and clear therefore the conclusions are reliable. Some minor revisions are needed before publishing.

1) Page 11-12, line 260-276, according to the data of Fig. 3E, the authors demonstrated the interaction of NyxA/NyxB with SENP3, and confirmed these interactions by NPM1. It will be better that adding the description of the comparison in the amount of NPM1 under the conditions of NyxA/NyxB and NyxA_MAG/NyxB_MAG, for better understanding to the readers.

2) Page 18, line 446-447, "Our work shows that the two proteins adopt the same ternary and tertiary structures", this sentence is not very rigorous. The crystal structure of NyxB was solved and confirmed by SAXS, however the structure of NyxA remains unknown. The amino acid sequences of two proteins are more than 80% identical. It means that the structure of NyxA is very possibly similar to NyxB but has not been verified experimentally.

3) The overall quality of crystallographic data is high but some over-refinement may exist. As shown in Supplementary Table 2, the RMS of bond angle of 1.12 degree is a little bit high for native data (2.5Å resolution). Also 0.07% of residues are located in Ramachandran outliers, it will be necessary to provide the density of these residues. Also the B-factor of solvent is smaller than the ones of macromolecules and ligands, it is abnormal. Some residues (~3% according to the PDB structure validation report) are poorly fit to the density maps. All these hints demonstrated the existence of over-refinement. The over-refinement should give some errors in the structure model but the conclusion related to crystal structure is still correct since the poorly fit residues are not the key residues for the functions discussed in this manuscript. However, for the rigor of the whole work, refinement of native data and update the Supplementary Table 2 are necessary.

Dear Referees,

Please find enclosed our point-by-point answer to all the reviewing comments. We thank the referees for the thorough reviews and excellent suggestions. We apologize for the delay in addressing them; due to multiple waves of COVID that hit our team, the technical challenges of working with a BSL3 pathogen, and serious technical issues reproducing the published ribophagy-related study presenting NUFIP as a specific receptor.

Reviewer #1 (Remarks to the Author):

Major points:

1. Although most of the data is strong and convincing, I am confused about the proposed overall mode of action of Nyx effectors on SENP3 with regard to Bifs and enhanced bacterial growth. While NyxA (and NyxB to a certain extent) causes mislocalisation of SENP3 and both clearly induce Bif formation in a manner dependent on binding to SENP3, do the authors claim that Nyx effectors retain SENP3 in the nucleus to prevent it from regulating ribophagy?

This is our current model, but further studies must be done to test this.

SENP3 has been reported to deSUMOylate multiple cytosolic targets in the context of stress, including Beclin1 in the case of starvation (Liu et al. 2020, DOI: 10.1080/15548627.2019.1647944) and more recently, mitochondrial fission protein 1 in the context of mitophagy (Waters et al. 2022, doi: 10.15252/embr.201948754). Given our results, we propose the following model: *Brucella* infection induces a cellular stress response that destabilizes the subcellular localization of nuclear proteins, notably their accumulation in cytosolic aggregates in the vicinity of replicating bacteria. This is enhanced by NyxA/B interaction with SENP3 that is retained in the nucleoplasm, promoting the formation of cytosolic aggregates enriched in ribosomal proteins. Bif formation partly depends on Beclin1, suggesting an autophagy-related process and contributes to efficient intracellular replication. A diagram has been included (Fig 6G), and the discussion has been modified accordingly to clarify our model. We also clearly state the questions that remain unanswered in the revised manuscript, notably the molecular function of these cytosolic structures and whether SENP3 retention prevents direct Beclin1 deSUMOylation.

If so, why are Nyx effectors detected in Bifs?

In infected cells, we detect 4HA-NyxA/B in Bif and the nucleus. In transfected cells, we also see NyxA/B in cytosolic and nuclear structures for HA-tagged Nyx effectors and, to a much lesser extent, 4HA-tagged Nyx effectors. This is clarified in the revised manuscript (Fig 1E and Supp Fig 2B, C). NyxA/B cytosolic accumulation may occur before nuclear import, and this phenomenon is artificially increased with the presence of bulkier tags such as the 4HA, which hampers efficient nuclear import (Supp Fig 2A). Alternatively, NyxA/B are cycling between these compartments. This will only be fully resolved when small tags can be used to detect translocated effectors during *Brucella* infection, ideally in live imaging. We are actively working on this as others in the field. Unfortunately, we cannot provide a technical solution to investigate the kinetics of the localization of Nyx during infection at this stage. We attempted live imaging of GFP-NyxA and NyxA-mCherry in transfected cells, but these tags reduce the nuclear import of Nyx, and fully inhibit accumulation in nuclear aggregates and give a diffuse nuclear localization (Fig A of this letter).

Figure A. Confocal image of HeLa cells expressing either HA-NyxA, GFP-NyxA or NyxA-MCherry. The presence of the bulky tags reduces nuclear import and prevents accumulation in nuclear aggregates.

Is SENP3 also recruited to Bifs, which are NyxA/B-induced structures?

No, SENP3 is not recruited to Bif. To address this question, we monitored cytosolic SENP3 using multiple approaches. We first carried out fractionation experiments and observed that the expression of NyxA/B in transfection does not induce a significant increase of cytosolic SENP3 (Fig B of this letter).

Figure B. HEK cells were transfected with either empty vector, HA-NyxA or HA-NyxB and were submitted to a fractionation procedure to obtain nuclear (NF) and cytosolic fractions (CF) as described in PMID: 27226300. The purity of each fraction was verified by monitoring the presence of tubulin (cytosolic) and fibrillarin (nuclear). SENP3 cytosolic and nuclear levels remain unchanged upon expression of either NyxA or NyxB.

Fractionation in infected cells is more challenging, as the percentage of infected cells is low even for HEK cells. We could not use bone marrow-derived macrophages because we could not find a suitable anti-SENP3 antibody for mouse cells. Therefore, we undertook a microscopy-based approach to enable the specific analysis of infected cells. Our usual immunofluorescence protocol does not reveal cytosolic SENP3 (Fig 4). We, therefore, used a different protocol involving cold fixation and very short strong permeabilization that has been reported to allow the detection of SENP3 in the cytosol during starvation (Liu et al. 2020, DOI: 10.1080/15548627.2019.1647944), a result that we confirmed upon extended EBSS treatment for 48h (Supp Fig 13). Although, using this protocol we could barely detect cytosolic SENP3 in *Brucella*-infected cells, in the few cells with detectable cytosolic SENP3 staining, we could not observe co-localization with translocated 3Flag-NyxA (Figure C).

$\Delta nyxAp3Flag-NyxA$

Figure C. A rare example of HeLa cells infected for 48h with $\Delta nyxAp3FLAG-NyxA$ exhibiting cytosolic SENP3-positive structures (white) which do not co-localize with 3FLAG-Nyx Bif. Scale bar corresponds to 5 μm .

We also attempted to treat infected cells with EBSS to see if NyxA/B retention of SENP3 would prevent its cytosolic accumulation, but EBSS starvation of infected cells resulted in their detachment. Finally, we carried out a proximity ligation assay to enhance the possibility of detecting weak signals. Although we could detect positive PLA between translocated 3FLAG-NyxA and NVL and RPL5, we could not detect any signal with SENP3 using our standard permeabilization or the cytosolic-specific protocol (Fig 5F, Supp Fig 12B). All this data is now included in the manuscript.

Consistently, we imaged transfected cells in which 4HA-NyxA was exclusively detected in the cytosol, and we observed an apparent reduction in the ability to delocalize SENP3 from the nucleoli (Fig D of this letter), suggesting Nyx nuclear localization is essential for this phenotype or the presence of the tag negatively impacts this phenotype. In the case of HA-NyxA transfected cells, we cannot fully dissociate the two locations as cells presenting cytosolic aggregates are almost always accompanied by nuclear staining, therefore strongly impacting SENP3 nucleolar localization.

Figure D. HeLa cells were transfected with either empty vector, HA-NyxA or 4HA-NyxA, for 13h. Selecting cells with exclusive cytosolic localization of 4HA-NyxA and majority cytosolic localization for HA-NyxA, we monitored the levels of co-localization between nucleolin and SENP3. The 4HA-NyxA has a significant loss in the ability to delocalize SENP3 from the nucleoli compared to HA-NyxA. Data are represented as means \pm 95% confidence intervals from 3 independent experiments. Each experiment is color coded and all events counted are shown.

In summary, we have not detected SENP3 in Bif with currently available tools. All our data suggest that SENP3 remains mostly nuclear during infection. Therefore, we propose a model in which nuclear NyxA/B partially retain SENP3 in the nucleoplasm, reducing its activity on the regulation of Beclin1-dependent stress responses initiated during *Brucella* infection. Whether this is due to lack of direct interaction with Beclin in the cytosol or an effect of decreased nucleolar localization remains to be determined.

In infected SENP3 KD cells, which form an increased number of Bifs, do NyxA and NyxB localise to the enhanced cytosolic structures?

We do not see a significant increase in the numbers of NyxA/B-positive structures in siSENP3 treated cells (Fig E of this letter; siControl: 14.4 ± 11.09 average 3FLAG-positive Bif versus 19.5 ± 10.14 average) suggesting depletion of SENP3 does not induce enhanced translocation of these effectors, nor their accumulation in these cytosolic structures. It also indicates that the formation of these structures is probably induced by *Brucella* infection, either as a stress response or due to the action of additional effectors yet to be identified. The function of NyxA/B seems to promote their formation but is not the initiation factor, as the expression of NyxA/B in transfected cells does not result in NVL/RPL5-positive Bif formation (Supplementary Fig 9C).

Figure E. Confocal image of HeLa cells infected with a mutant strain lacking NyxA but expressing 3FLAG-NyxA at 48h post-infection. Translocated 3FLAG was monitored (yellow), and bacteria/nuclei were visualised with Dapi.

2. The authors present evidence supporting a ribophagic nature of Bifs, in that they are positive for the ribophagy receptor NUFIP and their formation requires Beclin1. Yet, I think this important finding needs to be further substantiated with evidence for additional autophagic markers in Bifs: based on the data presented, is Beclin 1 detected on Bifs?

Given NUFIP presence on Bifs, is LC3B (Which NUFIP directly binds) detected?

Are lysosomal markers, such as LAMP1, present on Bifs, as would be expected for degradative ribophagy?

The lack of detection of ubiquitin using the FK2 antibody is interesting but not really pursued. It may reflect a ubiquitin-independent process or ribophagic structures that are matured enough for ubiquitin to be degraded.

To address this question, we have labelled Bif with either RPL5/NVL or with 3Flag-NyxA/4HA-NyxA and different markers suggested by the reviewer. We did not see any recruitment of LAMP1 (Supp Fig 17) for any of these combinations. LC3 was also not detected co-localizing with 3FLAG-positive Bif. However, significant proximity of Bif with large LC3-positive vacuoles was observed (Supp Fig 18A). Using transfected LC3-GFP to overcome methanol fixation, we did not observe substantial co-localization with either NVL or RPL5-positive Bif.

In the case of Beclin1, we could not successfully analyse its location by immunofluorescence microscopy. Three commercial antibodies were tested, and we only validated one of them by western blot with the siRNA depletion of Beclin1 (Table A of this letter, Supp Fig 16A). Unfortunately, this antibody is not compatible with immunofluorescence.

Table A. Antibodies tested for Beclin and NUFIP.

Antibody	Species	Source	Reference	Dilution IF	Fixation	Dilution WB	observation
Beclin	rabbit	abcam	ab207612	-	-	1/1000	validated with siRNA, but non functional in IF
Beclin	rabbit	abcam	ab62557	1-100	PFA	1/1000	Not validated with siRNA, IF looks like tubulin
Beclin	mouse	proteintech	66665-1-Ig	1/100	Antigenfix	1/10 000	Not validated with siRNA
NUFIP1	mouse	abnova	H00026747-B02P	1/100	PFA		staining still present in mKEIMA-RPS3 NUFIP1 KO
NUFIP1	rabbit	proteintech	12515-1-AP	1/50	PFA		staining still present in mKEIMA-RPS3 NUFIP1 KO

As an alternative, we transfected cells with Beclin1-HA. We mainly observed diffuse staining, with significant plasma membrane accumulation and no clear vesicular structures rendering image analysis of Beclin1 recruitment to Bif challenging. We could not see a strong enrichment of Beclin1 in Bif, but we observed a slight accumulation of Beclin1 surrounding some Bif (Figure F of this letter, arrowheads). However, more work is needed to confirm this. Because of the preliminary nature of this data, which we could not confirm with endogenous staining, we have chosen not to include it in the manuscript.

In summary, although we cannot exclude the presence of Beclin1 in Bif, our data suggest that Bif are not degradative autophagic compartments. This has now been included in the manuscript.

Figure F. Confocal image of HeLa cell infected with a mutant strain lacking NyxA but expressing 3FLAG-NyxA and transfected with HA-Beclin1 at 48h post-infection (transfection was carried out between 24 and 48h). Translocated 3FLAG was monitored (red) in relation to HA-Beclin1 detected with an anti-HA antibody (yellow) and bacteria/nuclei were visualized with Dapi.

Because of these results, we questioned whether the presence of NUFIP was potentially not related to ribophagy (Wyant et al. 2018, doi: 10.1126/science.aar2663) but its ribosomal biogenesis functions. We, therefore, revisited our NUFIP-ribophagy hypothesis and set out to measure the ribophagic flux in infected cells taking advantage of the RPL28-mKeima reporter cells kindly provided by Dr An Heeseon (Heeseon and Harper 2018, <https://doi.org/10.1038/s41556-017-0007-x>). We cannot use the microscopy-based readout, which would be the ideal method for analyzing infected cells, as it requires live imaging not available in our BSL3 facility. We relied on western blot detection of the Keima processed form, a much less sensitive technique considering we only have a small percentage of infected cells. No increase in RPL28-Keima processing was observed in *Brucella*-infected cells, suggesting there is not a significant induction of ribophagy (Figure G of this letter). However, in our hands, we could only detect Keima processing upon very high doses of arsenite treatment, in contrast to what has been reported (Heeseon and Harper 2018), so we may not be able to detect small changes that could occur during infection.

Figure G. Western blot of HEK293 RPL28 mKEIMA infected with either wild-type or a mutant strain lacking both NyxA and NyxB for 48h. Positive controls with 20 μ M and 1mM of arsenite were used as positive controls. Processed Keima (arrow) is indicative of the ribophagy flux.

Notably, Dr An in Professor Harper's team warned us that they failed to reproduce the role of NUFIP as a ribophagy receptor (An et al. 2020, doi:10.1038/s41586-020-2446-y; Extended data Fig 9) and cautioned us in the use of the anti-NUFIP antibody previously by Shim et al. 2020 (<https://doi.org/10.1080/15548627.2019.1662584>) to show NUFIP cytosolic accumulation. Using their NUFIP KO cells, we found a significant proportion of non-specific labeling was observed with the NUFIP antibody by microscopy (Supp Fig 15), making it unreliable. Given these results and considering the unclear role of NUFIP in controlling ribophagy, we have removed this data from our revised manuscript and modified the text accordingly. We have included a figure showing the non-specificity of the antibody labeling in the manuscript so that others do not waste time carrying out these experiments.

Altogether these data suggest that Bif are not degradative ribophagosomes. As their formation requires Beclin, suggesting an autophagy-derived process, we next analyzed if Bif could relate to ERphagy and Mitophagy processes by analyzing co-localization with the ER or mitochondria. Using 3Flag-NyxA and 4HA-NyxA as markers of Bif, we did not observe any co-localization with ER tracker (Supp Fig 19A) and only occasional labeling with mitotracker (Supp Fig 19B). It seems we are in the presence of a novel Beclin1-dependent compartment specifically induced during *Brucella* infection, enriched in ribosomal proteins. Further work is required to image and purify these structures to define their composition and function.

3. If Bifs represent ribophagy events providing nutrients to enhance bacterial growth, as speculated in the discussion, shouldn't SENP3 depletion also promote replication and not impair it, since it negatively regulates Beclin1-dependent ribophagy?
Or are there additional effects of SENP3 depletion to the cells that independently affect bacterial multiplication and confound the interpretation?

As SENP3 was previously shown to deSUMOylate Beclin1 in response to starvation (Liu et al. 2020, DOI: 10.1080/15548627.2019.1647944), nuclear detention of SENP3 would indeed prevent its ability to de-SUMOylate Beclin1 and reduce stress-induced autophagic events. However, retention of a proportion of SENP3 in the nucleus by the Nyx effectors at a specific stage of the infection is very different from the depletion of SENP3 throughout the whole infection (SENP3 inhibition takes several days to achieve and therefore is done a few days before the addition of the *Brucella* inoculum). Notably, SENP3 regulates ribosomal biogenesis (Haindl et al. 2008, doi: 10.1038/embor.2008.3), so its inhibition significantly reduces protein synthesis, potentially aggravating the lack of nutrients available for *Brucella*. This has been included in the discussion.

As our current model is that SENP3 cannot reach the host cytosol to de-SUMOylate Beclin1 and resolve *Brucella*-induced stress, we hypothesized that inhibition of the SUMOylation of

Beclin1 should have the reverse effect of depleting SENP3, and negatively impact the formation of Bif. As in starvation conditions, Beclin1 was shown to be SUMOylated by PIAS3 (Liu et al. 2020, DOI: 10.1080/15548627.2019.1647944), we depleted cells with siRNA (validation shown in Supp Fig 16B). We found that at 24h post-infection, the number of NVL-positive structures induced during infection is enhanced with the depletion of SENP3 and reduced with the depletion of Beclin1 and PIAS3 (Fig 6F). All this data is now included in the manuscript.

Also in this context, does Beclin1 KD affects optimal replication at 48 h pi?

We carried out replication counts at 48h post-infection by microscopy and observed a small but significant reduction in replication efficiency in cells depleted for Beclin1 and PIAS3. This data has now been included in the manuscript (Fig 6F), and the text has been modified accordingly, suggesting that Beclin1 and PIAS3 functions contribute to efficient intracellular replication at late stages of the infection in our experimental conditions.

Minor points:

1. Line 3: Based on the slight decrease in multiplication presented in Fig. 2C, it is a bit excessive to claim that SENP3 is

“essential” for intracellular replication. I suggest replacing it with “contributes to optimal growth” or an equivalent statement.

Similarly, I would tone down the sentence “...key role of SENP3 in the infectious process...” (lines 441-442).

Indeed this was an over-statement and the text has been modified.

2. Lines 85 and 455: The Brucella T4SS is characterised by the absence of the VirD4 component, so perhaps naming it “VirB T4SS” instead of “VirB/D” is more appropriate.

The sentence was corrected.

3. Line 92: ...with SENP3...

Corrected.

4. Fig. 1B-C. please add scale bars to the image and figure legend.

Apologies; scale bars were present but covered by the text. This has been corrected.

5. Fig 1E and S3A, lines 177-181: the pulldown experiments show a very small amount of the untagged proteins coming down in the eluates, so the interactions between purified Nyx proteins is not very convincing, despite the medium affinity K_D measured separately. There seems to be some methodological limitations to the interaction when using purified proteins. Could another technique be used, such as co-expression in cells followed by co-IP?

Ultimately, showing an interaction between the two effectors does not seem to add much to the manuscript, unless the authors suggest that heterodimerization is functionally meaningful. In the pull down experiments, the low amount of protein bound to its partner could be due to the presence of the tags, which may interfere with the efficiency of the interaction *in vitro*. This is why we performed the thermophoresis without tags. These show a clear interaction between NyxA and NyxB *in vitro* and a K_D of around 500 nM. We do not think testing their interaction *in cellulo* by over-expressing tagged versions of both proteins would help address this issue.

We agree with the reviewer that the interaction between the effectors does not add much at this stage to the role of NyxA/B in infection and have removed the data from the manuscript.

6. Lines 207-214: As above, given the limitations with obtaining purified full length SENP3, could a co-IP between Nyx and full length SENP3 in mammalian cells also confirm the interaction between these proteins? While not showing a direct interaction, it would complement the Y2H data convincingly.

The data showing the interaction between NyxA/B and SENP3 is: 1) the Y2H; 2) the interaction between purified NyxA/B and N-terminal part of SENP3; 3) the identification of the interaction domain, impacted with the mutation of the acidic groove; 3) recruitment of SENP3 in transfection, strongly reduced with the mutation of the acidic groove; and 4) the clear effect on SENP3 during infection also dependent on the acidic groove of Nyx. Nonetheless, we have carried out the proposed experiment. Endogenous SENP3 could not be used as it non-specifically binds myc and HA beads. Instead, we expressed GFP-SENP3, confirming its interaction with NyxA and NyxB using a GFP-trap (Fig 2B). The data has been included in the manuscript.

7. Fig. S8 and lines 349-353: Not to sound repetitive or obsessive, but the pull-down approach may not be the best methodology to use here, as an unusual cytosolic complex containing SENP3 and NVL may only form upon infection or ectopic expression of the Nyx effectors. A co-IPs of HA-Nyx effectors would be more appropriate in this instance and may reveal that SENP3 and NVL are part of a same Nyx-induced complex.

Ectopically expressed HA-Nyx recruits SENP3 but does not recruit NVL nor RPL5, which remain in the nucleoli (Supp Fig 9C). Therefore, NVL/RPL5 will not be part of the same complex as Nyx/SENP3 in transfection. Indeed, the cytosolic presence of NVL and RPL5 is induced upon *Brucella* infection, even in the absence of NyxA/B, whereas all our data suggest that SENP3 remains predominantly nuclear during infection. We attempted to do a co-IP of 4HA-NyxA in infected cells. However, we could not detect 4HA-Nyx in our HA trap, most likely due to the low amounts of translocated protein. To circumvent this, we carried out a proximity ligation assay (PLA) in infected cells. We observed a positive PLA signal between 3Flag-NyxA, NVL, and RPL5 but not SENP3 (using the cytosolic or our standard immunolabeling protocol). These results suggest that Nyx and NVL/RPL5 are less than 40 nm apart (DOI:10.1002/cpim.58) and may be part of the same complex in infected cells, at least transiently in Bif. These data have been included in the manuscript (Fig 5F).

8. Line 368: ...We hypothesised if... Please correct the grammar of this sentence.

The sentence has been modified.

9. Line 379: ...co-label NVL and RPL5....

The sentence has been modified.

10. I would suggest adding the quantification shown in Fig S13A to Fig 5F to emphasize this important piece of data.

This data has now been removed due to the non-specificity of the NUFIP antibody.

11. Lines 408-412: Please provide a Western blot of Beclin1 KD levels.

Apologies for this. The western blot is now included (Supp Fig 16A).

12. Line 415-417: the reasoning in this sentence is a bit awkward: one would expect a focus on SENP3 activity in Bifs, and not "elsewhere in the cell". Please clarify.

This has been modified.

13. Lines 506-508: This sentence implies that NyxA/B act in the nucleolus to retain SENP3 from regulating ribophagy in the cytoplasm, but they are detected in Bifs. This is linked to major point #1.

This has been explained in point 1 and the text has been modified to clarify.

14. Line 503: “overflow valve” could use a more scientific description.

We like using imagery to describe complex phenotypes, but the sentence has been removed.

15. Line 515: “Brucella could be eating the cell’s ribosomes” could be replaced with a more scientific description.

The sentence has been modified.

16. Lines 148 and 518: I would recommend using a more descriptive qualifier than “nicely”. Nicely has been replaced by well.

Reviewer #2 (Remarks to the Author):

In this manuscript, Louche et al. have demonstrated that Brucella effectors NyxA and NyxB interact with SENP3 in the nuclei, and these interactions seem to directly cause endogenous nucleolar SENP3 translocate to the nucleoplasm. However, deletion of both genes in Brucella does not seem to be sufficient to completely recover SENP3 nucleolar localization (suggesting the involvement of other unknown Brucella effector(s)). Moreover, SENP3 translocation of this kind also seemed to occur upon *B. abortus* infection, which also resulted in the cytoplasmic accumulation of the nucleolar protein NVL in Brucella-induced foci (Bif) in a manner dependent on Nyx effectors. SENP3 depletion seemed to affect *B. abortus* replication in HeLa cells, increase Bif, and induced the cytoplasmic accumulation of the nucleolar protein NVL in infected HeLa cells but not in non-infected controls. Interestingly the ribosomal protein RPL5, the Nyx effectors and the nucleo-cytoplasmic shuttling protein NUFIP1

known as a receptor for ribophagy were detected in the Brucella-induced NVL foci. (It is claimed that) Beclin1 knockdown seems to reduce Bif. Furthermore, *L. pneumophila* infection does not appear to result in cytoplasmic NVL accumulation so the phenomenon specifically occurs upon *B. abortus* infection. Although these observational findings are interesting and potentially important, the authors’ investigations are largely based on their speculations. Therefore, this study appears to be at an early stage, and substantial efforts are needed to figure out: i) if the mediating role(s) for SENP3 in these cellular events are dependent on its deSUMOylation activity; ii) if those proteins involved are deSUMOylating substrates for SENP3; iii) if SENP3-mediated autophagies including ribophagy or SENP3 mediated deSUMOylation of certain proteins results in Bif upon *B. abortus* infection.

We understand the reviewers’ comments and are aware that we have not fully elucidated the mechanisms involved. We have just been awarded a 4-year national grant to address many of these points, especially defining the role of SENP3 and SUMOylation during *Brucella* infection.

At this stage, we cannot address many of these comments due to technical challenges we first need to overcome, which hamper some of the suggested experiments. Firstly, the induction of the cytosolic accumulation of NVL and RPL5 only occurs during *Brucella* infection.

Therefore, the full effect of retention of SENP3 in the nucleoplasm rather than the nucleoli that NyxA/B mediates can only be assessed in infected cells and not in transfected cells.

Secondly, *Brucella* extensively replicates intracellularly but only infects a small proportion of cells (20-50% depending on the cell type). So, most of the signal detected in global approaches, such as biochemical assays and proteomics, will come from non-infected cells, often masking small specific changes. Lastly, *Brucella* is a high-security BSL3 pathogen that prevents us from carrying out many of the experiments we would like to do, notably sorting infected cells.

As it stands, this manuscript is intended to highlight two novel *Brucella* effectors that impact the subcellular localization of nuclear proteins during infection by targeting SENP3, whose function contributes to efficient intracellular multiplication. We also provide the structure of the effector and identify its interacting interface with SENP3, thereby defining an original hijacking mechanism. This is the first example of a bacterial pathogen impacting the spatial organization of nuclear proteins.

Given this reviewer's comments on the lack of mechanistic detail regarding the precise targets of SENP3 affected during infection, we have modified our text and toned down our conclusions.

My comments are listed as follows:

Major points

1) In the manuscript It has been speculated that *Brucella* effectors-SENP3 interaction negatively regulates deSUMOylation of Beclin-1 by SENP3 to cause NUFIP1-mediated ribophagy, in turn result in cytoplasmic accumulation of the nucleolar protein NVL in Bif that containing RPL5 and NUFIP1 upon *B. abortus* infection. To gain mechanistic insights as to how those cellular events really happen and how they are really associated, the following questions need to be answered:

Does expression of the two effectors or *B. abortus* infection increase in Beclin1 SUMOylation levels in the cells used?

Following experiments suggested by Reviewer 1, we have now excluded a role for ribophagy (see our answer to point 2 of reviewer 1 for all the information). Briefly, Dr An Heeseon in Professor Harper's team warned us that they failed to reproduce the role of NUFIP as a ribophagy receptor (An et al. 2020, DOI <https://doi.org/10.1038/s41586-020-2446-y>; Extended data Fig 9) and cautioned us in the use of the anti-NUFIP antibody to show NUFIP cytosolic accumulation. Using their NUFIP KO cells, we found a significant proportion of non-specific labelling was observed with the NUFIP antibody by microscopy (Supp Fig 15), making it unreliable. Given these results, the fact we could not detect a substantial increase in ribophagy during infection (Figure G of this letter) and considering the unclear role of NUFIP in controlling ribophagy, we have removed this data from our revised manuscript and modified the text accordingly. We have included a figure showing the non-specificity of the antibody labelling in the manuscript so that others do not waste time carrying out these experiments.

Nonetheless, Beclin1 is clearly implicated in the formation of cytosolic NVL/RPL5/Nyx-positive structures in response to *Brucella* infection. However, the expression of NyxA/B in transfected cells alone does not impact the localization of NVL/RPL5 (new Supp Fig 9C), despite efficiently retaining SENP3 outside nucleoli (Fig 4A-C), making this an inappropriate model to address the impact of these effectors on Beclin1.

We attempted to transfect a stable His-SUMO2/3 U2OS cell line (kindly provided by Prof. Ivo Hendricks) with HA-Beclin1 and then infect them with either wild-type *Brucella* or

mutant strains lacking NyxA/B for use in an HA-Trap experiment and subsequent anti-SUMO probing. However, when we controlled the samples by microscopy, less than 5% of HA-Beclin1-expressing cells were infected, rendering the experiment technically impossible. We do not currently have the means to sort infected cells in the BSL3 facility and analyze them specifically for SUMOylation.

Nonetheless, we carried out an additional experiment to strengthen this data. Beclin1 was shown to be SUMOylated by PIAS3 in the stress conditions, an interaction that is impaired by SENP3 (Liu et al. 2020, DOI: 10.1080/15548627.2019.1647944). We analyzed the formation of Bif in cells depleted for PIAS3 using siRNA (Supp Fig 16B). We observed a significant decrease in NVL cytosolic structures at 24h post-infection in cells depleted for PIAS3, equivalent to siBeclin1-depleted cells (Fig 6E). This is not observed in SENP3-depleted cells, which show more NVL-positive structures (Fig 6B, C, D and E). Together this data suggests that Beclin1 and PIAS3 induce the formation of these structures whereas SENP3 negatively regulates them. Significantly, the depletion of PIAS3 and Beclin1 impacted *Brucella* intracellular replication at the late stages of the infection (Fig 6F), suggesting their functions are important for pathogenesis. All this data is now included in the manuscript. We propose that retention of SENP3 in the nucleus by NyxA/B has an effect on the function of Beclin1 that would modulate the formation of these cytosolic structures as a stress response to *Brucella* infection. As we did not succeed in determining the levels of Beclin1 SUMOylation in infected cells, we have clearly stated so in the discussion and the represented model (Fig 6G).

Does SENP3 deSUMOylate Beclin-1 in non-infected cells?

It has been reported that SENP3 de-SUMOylates Beclin1 (SUMO3 conjugation) in starvation conditions (Liu et al 2020, DOI: 10.1080/15548627.2019.1647944).

Does *B. abortus* infection causes general autophagy and/or selective autophagy types, including ribophagy (by conducting LC3 conversion and ribophagic reporter assays (1)), through regulating SUMOylation status of Beclin1?

It has been shown that *Brucella* modulates autophagy at different steps of the infection. Several lines of evidence support selective autophagy's role in the *Brucella* intracellular life cycle. Atg9 and WIP1 have been implicated in forming the replicative vacuoles (Taguchi et al., 2015 PMID: 25742138), and transient multi-membrane *Brucella* vacuoles with monodansyl cadaverine have been described in epithelial cells (Pizarro-Cerdá et al., 1998a; 1998b PMID: 9573138; PMID: 9826346). However, *Brucella* vacuoles are devoid of the classical autophagy marker LC3 suggesting components of the late steps of autophagy are not implicated. Importantly, for the last stage of the *Brucella* intracellular trafficking, ULK and Beclin1 have been implicated, mediating bacterial exit from infected cells (Starr et al., 2011 PMID: 22264511). More recently, it was shown that extensive replication of *Brucella* induces BNL3-mediated mitophagy, impacting the exit of bacteria from infected cells (BioRxiv: doi: <https://doi.org/10.1101/2022.08.31.505824>).

We have now carried out experiments to precisely monitor the ribophagy flux, and we could not detect any significant impact by *Brucella* (Fig G of this letter). However, this could be to the low percentage of infected cells. In addition, no LC3 nor LAMP1 were detected in Bif, suggesting these are not degradative autophagosomes (Supp Fig 17 and 18). These are also negative for endoplasmic reticulum and mitochondrial markers (Supp Fig 19). Therefore, we may be observing a selective highjacking of autophagy machinery by *Brucella* resulting in the accumulation of ribosomal proteins in the vicinity of replicating bacteria.

Does expressing the effectors or *B. abortus* infection cause changes in Beclin1 expression levels in the cells used?

We have carried out this experiment as suggested and could not detect any significant changes in Beclin1 levels after 48h of infection (Figure H of this letter). These experiments were done in bone marrow-derived macrophages (iBMDM), with the highest infection rate for *Brucella* (approximately 50%).

Figure H. Immortalized BMDM were infected for 48h with either wild-type or a mutant strain lacking both NyxA and NyxB. Cells were lysed, and the levels of Beclin1 were assessed by western blot using an anti-Beclin1 antibody. No significant differences were observed in comparison to mock-infected cells (NI). Samples from 2 independent experiments are shown (labelled 1 and 2).

Is NUFIP1 involved in *B. abortus*-induced Beclin1-dependent ribophagy (if any)?

As explained above, we have removed the NUFIP data from the revised manuscript.

Is Bif resulted from *B. abortus*-induced Beclin1-dependent ribophagy (if any)?

We have now carried measurements of ribophagy flux and have not detected any induction during *Brucella* infection (Figure G of this letter). In addition, Bif lack LC3 and LAMP1. We have, for the time, excluded ribophagy. However, the formation of Bif requires Beclin1 and PIAS3 activities. At this stage, we can only speculate that these are a new stress response compartment induced by *Brucella* infection.

2) As a potent autophagy inducer, Rapamycin (originally isolated from bacterium *Streptomyces hygroscopicus*) is known to induce nucleolar SENP3 translocate to the nucleoplasm via inhibiting mTOR pathway. It would be necessary to know if expression of the two effectors or *B. abortus* infection affects mTOR signaling.

We have not tested the role of mTOR as all our data support NyxA/B retention of SENP3 by direct binding. However, it is possible that NyxA/B binding to the N-terminus of SENP3 interferes with NPM1 interaction (although its localization is not affected) and/or mTOR phosphorylation, which would interfere with nucleolar shuttling. We have not tested this possibility.

3) NVL, RPL5 and NUFIP1 are identified as SUMO-2 target proteins identified by mass spectrometry (2): K208, K164/K188/K221, and K229 are SUMO conjugation sites in NVL, RPL5, and NUFIP1, respectively. Therefore sequestering or depletion of SENP3 would enhance their SUMOylation and that would result in their translocation(s).

In this respect, the following questions need to be answered:

Does expression of the two effectors or *B. abortus* infection increase in SUMOylation levels of these 3 proteins in the cells used? Does SENP3 depletion increase in SUMOylation levels of these 3 proteins in the cells used? Does SUMOylation regulate the translocation of NVL, RPL5 or NUFIP1? Does sequestering/depletion of SENP3 initiate the occurrence of multiple SUMO conjugations simultaneously at spatially related groups of target proteins such as NVL, RPL5, NUFIP1 and possibly Beclin-1 to induce autophagy/ ribophagy upon *B. abortus* infection

These are all excellent questions we will have to address in the future. We must first overcome our current technical challenge to sort infected from non-infected cells. Once we

achieve this, we will be able to carry out a full SUMOylome on infected cells in the presence or absence of the Nyx effectors. This is all part of the grant that we have been awarded and we anticipate at least 1 year to carry out these experiments. Since we cannot currently define the precise targets of SENP3 impacted by the presence of NyxA/B, we have clearly stated this in the manuscript and modified our conclusions.

4) RPL5 gene seems to be regulated by SUMOylation (3). Therefore, it is necessary to examine if *B. abortus* infection induces changes in the levels of free SUMOs and SUMO conjugation in the cells used in this study.

We have tested the levels of SUMOylated proteins in infected cells as suggested. No overall changes can be detected, suggesting that only a small proportion of the targets of SENP3 may be impacted (Fig I of this letter). This is in contrast to what has been described for *Listeria* that targets Ubc9 resulting in a severe reduction in the overall levels of SUMOylated proteins during infection (Ribet et al. 2010 PMID: 32350466). In our case, the lack of a major effect on overall SUMOylated proteins is not surprising since NyxA and NyxB retain part of SENP3 at a specific stage of the infection but do not fully deplete cells of this deSUMOylase. It is also possible that we cannot detect these changes due to the low percentage of infected cells.

Figure I. Immortalized BMDM were infected for 48h with either wild-type or a mutant strain lacking both NyxA and NyxB. Cells were lysed and the levels of SUMOylated proteins were assessed by western blot using anti-SUMO1 or SUMO2/3 antibodies. No significant differences were observed in comparison to mock-infected cells (NI). A positive control of cells treated with Arsenic 5 μ M for 3h was included. Anti-histone labelling was used as a loading control.

5) Lack of mechanistic explanation regarding the discrepancy in subcellular locations of the Nyx receptors differentially tagged (i.e., HA/Myc vs 4HA/3Flag).

The mechanisms of nuclear import of these effectors remain to be discovered. They do not possess any typical nuclear localization signals and may be likely “piggyback” riding by associating with another protein. Defining the mechanism of nuclear import is, in our opinion, out of the scope of this manuscript.

6) Lack of mechanistic explanation why SENP3 depletion increased the appearance of cells with “1-5” and “6-10” bacteria 48 h post infection (Figure 2C).

To determine the impact on intracellular replication, we divided infected cells into 3 categories: no/low replication (1-5 bacteria), medium replication (6-10 bacteria) and high replication (>10 bacteria). The graph shows the percentage of cells in each category, totaling 100%. If there is a negative impact on the number of cells with heavily replicating bacterial loads (>10), the other categories become more represented, increasing their percentages. As *Brucella* replicates in normal cells, one observes a clear decrease of 1-5 and 6-10 bacteria, accompanied by increased cells with more than 10 bacteria.

7) Lack of mechanistic explanation regarding why “NyxA did not form crystals in any of the conditions tested” for solving its structure.

This sentence has been removed from the manuscript. There is no mechanistic explanation for crystal formation at this stage. One possibility is that the slight differences in the N-terminal portion of the protein stabilize crystal formation.

8) Lack of mechanistic explanation regarding the observation that the authors stated on the page 13: “we could also observe a statistically significant increase of SENP3 in the nucleoli in cells infected with the Δ nyxB strain compared to cells infected with wild-type *B. abortus*, although to a lesser extent than what we observed for Δ nyxA.”

In all our experiments, NyxB has a lesser striking phenotype than NyxA. NyxB may have a lower affinity for SENP3 *in cellulo*, which is currently impossible to test. Alternatively, additional functions or partners may be targeted by these effectors that we have not yet discovered, which will differentiate NyxA and NyxB activities. Both effectors are likely important for *Brucella* pathogenesis as they have both been maintained through evolution.

9) Non-infected controls are needed for Figure 5A and Supplementary Figure 9; These have been added (Fig 5A and C; Supp Fig 7A and B).

10) On the page 7, experimental evidence is needed for the authors’ statement: “This hypothesis was not confirmed in infected cells as effectors tagged with a single HA epitope were not detectable at any of the time-points analysed.”

An example of imaging of cells infected with *Brucella* expressing HA-NyxA, in which no signal is detected apart from weak background staining, is now included (Supp Fig 2D).

11) On the page 9, amino acid sequence details are needed for the authors’ statement: “This region encodes two basic amino acid stretches predicted as nucleolar localisation sequences...”

We apologize for our lack of clarity. The NoLS were predicted by the NoD server (Scott MS et al., Troshin PV, Barton GJ. BMC Bioinformatics. PMID: 21812952). The region is also involved in NPM1 binding and mTOR phosphorylation that was beautifully implicated in the nucleolar shuttling of SENP3 in the study of Raman *et al.* (*Mol Cell Biol* **34**, 4474–4484 (2014) PMID: 25288641). The text was modified accordingly.

See Figure J of this letter.

23 AYSSPRRERLRWPPPKPRLKSGGIGFGDPGSGTTVPARRLPVPRPSFDASASEEEEEEEEEDEEEEEV
AAWRLPPRWSQLGTSQRPRPSRPTHKTCQRRRRAMRAFRMLLYSKSTSLTFHWKLGWRHGR 159

Figure J. N-terminal sequence (residues 23 to 159) of SENP3 with lysine and arginine residues indicated in bold blue and mTOR phosphorylation sites in red. The region involved in NPM1 binding is shaded in yellow and the predicted nucleolar localisation signals are indicated by a dashed box.

12) On the page 11, the authors stated, “we could only detect a small amount of endogenous SENP3 in the cell extract with our antibody”. According to the Supplementary Information, the antibody refers to the one (D20A10) from Cell Signaling Technology. Is possible to use an alternative antibody to further verify the finding?

This was the only antibody tested we could validate for microscopy with siRNA. Instead, we have carried out a co-immunoprecipitation with GFP-SENP3 to confirm the interaction *in cellulo* (Fig. 2B).

13) Lack of sequence details of non-targeting siRNA as well as siRNA targeting SENP3 or Beclin1: they do not appear to be included in the Supplementary Table 6.

Apologies for the omission; these have now been included in the table.

14) No experimental evidence showing Beclin1 siRNA-mediated depletion (either western blotting or qPCR confirmation is needed.)

This has now been included for Beclin1, the new inhibition of PIAS3 (Supp Fig 16A and B), and a second siRNA for siSENP3 (microscopy and western blot; Fig 2E).

15) On the page 16, the authors stated, “Due to antibody incompatibility we could not co-label with NVL and RPL5 simultaneously”. The supplementary table 7 used two antibodies (rabbit and mouse origins, respectively) for NVL detection and one rabbit antibody for RPL5. So the incompatibility needs to be clarified and an alternative experiment needs to be conducted.

Apologies for the confusion. The mouse antibody was used only for western blotting (not in the paper) and did not work in immunofluorescence microscopy with our protocol. The Table has been corrected.

16) On the page 18, the authors stated, “infection with *L. pneumophila* did not result in NVL cytoplasmic accumulation”. Is it possible to introduce the Nyx effectors into the *L. pneumophila* to see if the bacteria can induce the appearance of NVL in the cytoplasm of infected cells in the presence of the Nyx effectors.

This would be excellent. Unfortunately, we have never succeeded in secreting any effectors via the type IV secretion system of *Legionella*. These are two different systems: *Legionella* is classified as a type IVB secretion system, whereas *Brucella* is a type IVA. The loading mechanisms are not conserved between these two systems.

17) On the page 20, the authors demonstrate their siRNA-mediate SENP3 depletion (Figure 2B). However they stated, “Full depletion of SENP3...”. Actually in this study, no knockout technology was used to fully deplete SENP3. So the statement is scientifically inaccurate.

This has been corrected; apologies for the poor choice of words.

Crucially to exclude potential siRNA off-target effect(s), it is necessary to use at least another siRNA to knock SENP3 down for verifying those effects observed in this study.

We have done this as requested (Fig 2E), and all data is included in the revised manuscript. We observe the same effect on the formation of NVL-positive structures (Fig 6E) and intracellular replication (Fig 2F).

Furthermore, to investigate if SENP3 action upon *B. abortus* infection is dependent on its deSUMOylation activity, SENP3 knockdown plus rescue experiments (with SENP3 wild type protein or inactive SENP3 mutant) should be essential in the cells used in this study.

Again, this is an excellent idea. However, after depletion, when we subject cells to transfection for the rescue and subsequent infection, we have no cells left for analysis.

Therefore, the experiment is currently not possible with the available tools; CRISPR SENP3 mutants will have to be engineered, something we are now doing in the lab and will provide to the community when validated.

Minor points

1) References are needed for:

The statement on the page 13, “One of the principal roles of SENP3 in the nucleoli is to regulate ribosomal biogenesis, specifically of the 60S ribosomal subunit.”

The statement on the page 15, “we infected immortalised bone marrow-derived macrophages (iBMDMs), a well-established model of Brucella infection”.

All have been added.

2) Few grammatical errors were spotted:

On the page 15, “if” (line 368) should be “that”

On the page 18, “that” (line 440) should be “which”

On the page 20, the sentence, “siRNA depletion of SENP3 occurs at an earlier stage of the replication than the effect induced upon NyxA/B translocation” (lines 484 and 485), is not easy for understanding. Please revise it.

On the page 22, “Alternative” (line 543) should be “Alternatively”

All have been corrected.

Reviewer #3 (Remarks to the Author):

The MS of A. Louche et al. reports the identification of NyxA and NyxB, two new Brucella effectors that are translocated to the host cell where they target the SENP3 protease, promoting its delocalization from the nucleoli. These new effectors promote the formation of cytoplasmic structures enriched in the nucleolar proteins NVL, NUFIP1 and RPL5, in a process that is dependent on Beclin1 and negatively regulated by SENP3. The authors propose that NyxA/B, acting as two nucleomodulins, interact with SENP3 to enhance a ribophagy-like process that ultimately promotes Brucella intracellular proliferation. This is an original, extensive, and relevant piece of work describing what seems a new family of bacterial effectors with nucleomodulatory functions. The authors succeed in identifying both effectors and their cellular target, which is per se a great task. However, they went beyond and resolved the crystal structure of NyxB what allow them to describe the acidic groove necessary for the interaction with SENP3, thus providing the biophysical basis of the interaction and confirming the Y2H and immunoprecipitation experiments. Then, they studied the effects of NyxA/B-SENP3 interaction discovering that it also promotes de delocalization of NVL and RPL5 from the nucleoli, accumulating in cytoplasmic structures called Brucella induced foci (Bif) where the effectors are also located. The ribophagy receptor NUFIP1 is also recruited to the Bif in a NyxA/B dependent manner. So, the authors propose that NyxA/B target SENP3 to enhance cellular ribophagy as a way to promote the Brucella intracellular proliferation.

The MS is well written and clear for the reader.. The main body contains the most important figures of the work, and the supplementary material is abundant and provides the necessary information to fully underst the article. The experiments are well designed and conducted in an elegant manner. In summary, this is an excellent and stimulating paper providing compelling evidence about a new family of bacterial effector with nucleomodulatory functions.

Minor comments:

Ln114. Ref 19 reports a Legionella effector not a “subgroup pf Brucella candidate effectors”. Please revised it.

This publication is correct, as it lists the candidate *Brucella* effectors in Table S1.

Ln162-183. Does the authors have any hypotheses as to why NyxA and NyxB interact? Is this

interaction important for targeting SENP3? Can they form a heterodimer? I would like to know what ideas the authors have about this.

We do not think the interaction is important for targeting SENP3 as they can efficiently interact with SENP3 individually.

We have not detected the formation of a heterodimer *in vitro*. The crystal structures suggest instead that the two dimers are fully compatible and we model a potential heterotetramer as depicted in Figure K of this letter. Our current hypothesis is that there is the formation of an heterotetramer. This would be consistent with the low affinity between them. Experiments are ongoing to confirm this by different biochemical approaches, mutating both the dimer and potential interaction interfaces, and test their role in infection. As suggested by reviewer 1, we have removed the interaction data from this manuscript. We will pursue the characterization of its function in the context of infection to ensure this interaction is relevant in infection and *in cellulo* rather than simply *in vitro*.

Figure K. Model of potential interaction between NyxA and NyxB dimers, in green and orange respectively.

Reviewer #4 (Remarks to the Author):

This manuscript presents a great job and is very worthy of publication. The author identified two new *Brucella* effectors, NyxA and NyxB, confirmed the target of these two effectors as SENP3 and discussed the functions during infection. The finding is novel and very helpful for understanding the mechanism of pathogen infections. The data as well as the explanations are solid and clear therefore the conclusions are reliable. Some minor revisions are needed before publishing.

1) Page 11-12, line 260-276, according to the data of Fig. 3E, the authors demonstrated the interaction of NyxA/NyxB with SENP3, and confirmed these interactions by NPM1. It will be better to add the description of the comparison in the amount of NPM1 under the conditions of NyxA/NyxB and NyxA_MAG/NyxB_MAG, for better understanding of the readers.

We have now modified this.

2) Page 18, line 446-447, “Our work shows that the two proteins adopt the same ternary and tertiary structures”, this sentence is not very rigorous. The crystal structure of NyxB was solved and confirmed by SAXS, however the structure of NyxA remains unknown. The amino acid sequences of two proteins are more than 80% identical. It means that the structure of NyxA is very possibly similar to NyxB but has not been verified experimentally.

We agree that our sentence is not rigorous and have modified the text accordingly by replacing the sentence with: “Given the high sequence identity between the two proteins and their structural similarity in SAXS, it is likely that the two proteins adopt very similar ternary and tertiary structures”

3) The overall quality of crystallographic data is high but some over-refinement may exist. We agree that the statistics of the refined structure could indicate over-refinement. So we have reprocessed the data and refined the structure again with statistics now presented in Supplementary Table 2.

As shown in Supplementary Table 2, the RMS of bond angle of 1.12 degree is a little bit high for native data (2.5Å resolution).

The new RMS of the bond angle is 1.33.

Also 0.07% of residues are located in Ramachandran outliers, it will be necessary to provide the density of these residues.

This number is now 0.

Also the B-factor of solvent is smaller than the ones of macromolecules and ligands, it is abnormal.

We thank reviewer 4 for this interesting comment that, admittedly, puzzled us. At first, we agreed with the reviewer, but after a thorough examination of the literature and our reprocessed data, we do not think it is not abnormal; perhaps, even more, the contrary. We have looked into the recent crystal structures solved and published in various journals.

There we found many examples where waters B factors were lower than protein/ligands (see below four examples and references at similar resolutions).

Article; PDB code; Bfactor waters; Bfactor macromolecules

Gao et al., 2021 Nature communication ; 7DHG ; 56.75 ;59.03

Morris et al., 2021 Nature communication ; 7R8X ; 53.79 ; 54.51

Guegueniat et al., 2021 Nucleic Acid Research; 5KKP; 52.1; 59.7

Lipper et al., 2021 Structure ; 7RDB ; 69.06 ; 95.80

We believe that this likely reflects that density is clear only for the most stable waters and thus with low B factors. In our new NyxB structure that we have re-deposited (PDB code 8AF9), the B factors are now :

Protein: 52.84

Waters: 48.16

Some residues (~3% according to the PDB structure validation report) are poorly fit to the density maps.

This number is 1%.

All these hints demonstrated the existence of over-refinement. The over-refinement should give some errors in the structure model, but the conclusion related to crystal structure is still correct since the poorly fit residues are not the key residues for the functions discussed in this manuscript. However, for the rigor of the whole work, refinement of native data and update the Supplementary Table 2 are necessary.

We now present a new Supplementary Table with statistics reflecting the modifications. The Validation report is also included annexed to this letter.

Reviewer #1 (Remarks to the Author):

In this manuscript, Louche and colleagues present a revised version of a study describing the discovery, structure and mode of action of novel effector proteins delivered into host cells by the intracellular pathogen *Brucella abortus*, which act via sequestration of the SUMO protease SENP3 and relocalisation of nucleolar proteins to the host cell cytoplasm into structures whose formation depend upon the autophagy initiation protein Beclin1 and the SUMO-E3 ligase PIAS3, a process that may promote bacterial intracellular proliferation.

The authors should be commended for their extensive efforts at addressing all reviewers' comments, which have significantly improved the manuscript and also addressed my own concerns in a satisfying manner. I appreciate in particular that the authors have uncovered unreliable reagents that they originally used to invoke ribophagy in NyxA/B-dependent Bif formation and have now removed these results. Although the nature and function of Bifs remain mysterious, this manuscript is further completed by the findings of the roles of Beclin1 and PIAS3 in Bif formation and overall constitutes an extensive characterization of unique bacterial effector proteins of importance to intracellular pathogenesis and with new nucleomodulatory modes of action.

Below are some minor, text-editing requests.

1. line 412: should be "revealed"
2. line 564: "...which SUMOylates..."
3. line 583: should be "hijacking"

Reviewer #2 (Remarks to the Author):

It is evident that the authors have conducted substantial further investigation with significant new data included, which have substantiated their discovery regarding the two novel *Brucella* effectors that regulate the subcellular localisation of nuclear proteins through targeting SENP3 upon infection. Interestingly and importantly, these findings also suggest a potential novel mechanism where *Brucella* selectively hijacks host autophagy machinery in promoting ribosomal protein accumulation around replicating bacteria. As a result of their great efforts, following the revision, the quality of this manuscript has been significantly improved.

Moreover, I greatly appreciate the technical challenges that the authors have encountered during their attempts to address my questions/concerns. Many congratulations on their 4-year grant newly awarded. I am looking forward to reading more about their new findings in future in this direction.

Furthermore, there is one more small thing that I would like the authors consider to make changes:

On the page 23, between lines 563-565, there seems a grammatical error in the complex sentences: "This is consistent with the requirement of the SUMO E3 ligase PIAS3 for the formation of Bif, SUMOylates Beclin during starvation, a process negatively regulated by SENP3." For clarity it would be better to rewrite the sentences.

Reviewer #4 (Remarks to the Author):

I am very glad to see the authors re-processed the diffraction data and better crystal structures were obtained to support the finding. For the question of B factors, I agree that if only stable waters with clear densities were selected, the B factors would be similar, even smaller than the ones of proteins and ligands. In fact, the new number of B factors are 52.84 (protein) and 48.16 (water), which are almost same values, since the errors of B factors are bigger than other parameters, such as the positions of atoms.

I agree to publish this manuscript.

Dear Referees,

We thank you for the text-edit suggestions. We have included all suggestions in the revised manuscript.

Reviewer #1 (Remarks to the Author):

Below are some minor, text-editing requests.

1. line 412: should be “revealed”
2. line 564: “...which SUMOylates...”
3. line 583: should be “hijacking”

These were all modified.

Reviewer #2 (Remarks to the Author):

On the page 23, between lines 563-565, there seems a grammatical error in the complex sentences: "This is consistent with the requirement of the SUMO E3 ligase PIAS3 for the formation of Bif, SUMOylates Beclin during starvation, a process negatively regulated by SENP3." For clarity it would be better to rewrite the sentences.

This was modified as follows:

This is consistent with the requirement of the SUMO E3 ligase PIAS3 for the formation of Bif, which SUMOylates Beclin during starvation, a process negatively regulated by SENP3⁵.